# Secondary ice production during the break-up of freezing water drops on impact with ice particles

Rachel L. James[1], Vaughan T. J. Phillips[2], and Paul J. Connolly[1]

[1]Department of Earth and Environmental Science, The University of Manchester, Manchester, UK
[2]Department of Physical Geography and Ecosystem Science, University of Lund, Lund, Sweden

**Correspondence:** Rachel L. James (rachel.james@manchester.ac.uk)

**Abstract.** We provide the first dedicated laboratory study of collisions of supercooled water drops with ice particles as a secondary ice production mechanism. We experimentally investigated collisions of supercooled water drops ($\sim$5 mm in diameter) with ice particles of a similar size ($\sim$6 mm in diameter) placed on a glass slide at temperatures > -12 °C. Our results showed that secondary drops were generated during both the spreading and retraction phase of the supercooled water drop impact. The secondary drops generated during the spreading phase were emitted too fast to quantify. However, quantification of the secondary drops generated during the retraction phase with diameters > 0.1 mm showed that 5–10 secondary drops formed per collision, with approximately 30% of the secondary drops freezing over a temperature range between -4 °C to -12 °C. Our results suggest that this secondary ice production mechanism may be significant for ice formation in atmospheric clouds containing large supercooled drops and ice particles.

## 1 Introduction

Most surface rainfall events that occur across the globe are associated with the ice–phase within clouds in the Earth's atmosphere (Field and Heymsfield, 2015), as are severe weather events such as freezing rain, hail and thunderstorms (Changnon, 2003; Púčik et al., 2019; Elsom, 2001). Therefore, understanding the processes which govern ice formation in clouds is crucial for determining their effects on both climate and weather.

Where subzero temperatures are warmer than the homogeneous freezing point of -35 °C, supercooled water drops can heterogeneously freeze via a subset of aerosol particles present in the atmosphere. This subset of aerosol particles, called ice nucleating particles (INPs), are relatively rare, and while number concentrations of INPs vary in time and space, they typically fall between $1 \times 10^{-5}$ L$^{-1}$ to $1$ L$^{-1}$ at $\sim$-10 °C (Kanji et al., 2017). Yet, observed ice particle concentrations in mixed–phase clouds can be several orders of magnitude higher than concentrations predicted from ice particles forming due to INPs (e.g. Crawford et al., 2012; Lloyd et al., 2015; Lasher-Trapp et al., 2016; Ladino et al., 2017). Ice can also form at temperatures > -35 °C via secondary ice production (SIP) where new ice particles are formed from pre–existing ice particles. However, our understanding of ice formation from SIP mechanisms is incomplete (e.g. see reviews by Field et al., 2017; Korolev and Leisner, 2020), resulting in poor representation of SIP mechanisms in numerical weather prediction (NWP) models.

Observations within mixed–phase clouds often show ice crystal number concentrations higher than the numbers of ice nu-
25 cleating particles present in the atmosphere. For instance, ice particle number concentrations exceeding 100 L$^{-1}$, in shallow
convection with cloud–top temperature no lower than -12 °C, have been observed over the UK (Crawford et al., 2012). Fur-
thermore, thin mixed–phase layer clouds have been observed to continually generate snow (Westbrook and Illingworth, 2013).
Conventional thinking would suggest that the ice in mixed–phase layer clouds should fall out, leaving the layer 'depleted' of
INPs; however, the observations clearly show that ice continues to form in these clouds over time.

The rime–splintering SIP mechanism has been successful in predicting the glaciation of mixed–phase clouds in many cases,
especially those involving a warm cloud base creating sufficiently large cloud-drops in the rime-splintering temperature region
between -3 °C to -8 °C (e.g. Harris-Hobbs and Cooper, 1987; Blyth and Latham, 1993, 1997; Phillips et al., 2001, 2005;
Crosier et al., 2011; Crawford et al., 2012; Taylor et al., 2016; Huang et al., 2017). However, there are also numerous cases
where significant concentrations of ice observed in clouds cannot be explained by the rime–splintering SIP mechanism. Hobbs
and Rangno (1985) compiled tables of aircraft observations from a wide range of cloud environments. They found that the max-
imum ice particle concentrations were independent of the cloud–top temperature but were strongly dependent on the broadness
of the supercooled drop spectrum near the cloud–top, with approximately half of the clouds exhibiting ice enhancement.

Several SIP mechanisms have been identified and studied both in the laboratory and theoretically, but only the rime-
splintering SIP mechanism is widely implemented in NWP models. Active between -3 °C to -8 °C, rime–splintering occurs
when supercooled water drop diameters are < 13 μm and > 24 μm (Hallett and Mossop, 1974; Mossop and Hallett, 1974;
Mossop, 1978). Another SIP mechanism, the fragmentation of freezing drops, has received a significant proportion of labo-
ratory based SIP investigations. Fragmentation due to freezing drizzle drops or raindrops can occur over a wider temperature
range between 0 °C to -32 °C, but quantification of ice fragment generation rates and temperature dependence within these
rates between laboratory studies varies significantly (see Table 1 of Korolev and Leisner, 2020, for a summary). A range in
diameters of freezing supercooled water drops has also been investigated between laboratory studies from 4 μm to 1000μm (see
Table 1 of Korolev and Leisner, 2020, for a summary). While other SIP mechanisms exist (e.g. ice–ice collisions, sublimation
fragmentation), the attention of laboratory studies has overwhelmingly focussed on the SIP mechanisms of rime–splintering
and fragmentation due to freezing drops. Furthermore, unidentified SIP mechanisms may also exist.

In this paper, we present a SIP mechanism involving the formation of secondary drops from the collision of a supercooled
water drop with a larger ice particle. This SIP mechanism has been investigated via a theoretical study by Phillips et al. (2018)
who referred to it as 'Mode 2' as it involves collisions of supercooled water drops with more massive ice particles resulting in
fragmentation of the supercooled water drop. Ice contained in some of the secondary drops was assumed to initiate freezing,
yielding secondary ice fragments. By contrast, 'Mode 1' involved either collisions of supercooled water drops with less massive
ice particles resulting in spherical freezing of the supercooled water drop or activation of immersed INPs, with a quasi-spherical
outer ice shell that fragments.

While there are no dedicated laboratory studies of this SIP mechanism involving collisions of supercooled water drops with
more massive ice particles or activation of INPs immersed in them, there are laboratory studies that have indirectly studied as-
pects of this process. For example, a similar mechanism was alluded to by Latham and Warwicker (1980) in their experimental

investigation of charge transfer during interactions between hailstones and supercooled water drops. They observed that frost could occasionally be broken during impact, thus forming new ice particles. Although this was an unwanted outcome of their experiments it provided some hints of a potential SIP mechanism during the interactions between ice particles and supercooled raindrops. Later, Schremb et al. (2018) studied the fluid flow and solidification of supercooled water drops on elevated ice targets, briefly observing the formation of secondary drops from the rim of the supercooled water drop during impact. However, for both of these studies no quantification of the secondary drops was made.

In this paper, we describe a set of experiments performed at the University of Manchester to determine the freezing fraction of secondary drops ($\Phi$) formed in the splash during the collision of 5 mm diameter supercooled raindrop on a 6 mm diameter ice particle, providing the first laboratory quantification of this SIP mechanism. This freezing fraction ($\Phi$) is the ratio of secondary drops that freeze to all such drops emitted. The experimental setup is described in Section 2. The results are presented in Section 3, and the discussion in Section 4. Finally, the conclusions are given in Section 5.

## 2 Experimental Setup

A schematic of the experimental setup is shown in Fig. 1. The setup was purpose–built to study the impact of a supercooled water drop on an ice particle. For this study, we used two configurations of the experimental setup. The first configuration was used to study the drop impact with a high–speed camera (Chronos 1.4, Kron Technologies Inc.) equipped with a microscopic lens (Kron Technologies Inc.) and a 0.5× barlow lens (Kron Technologies Inc.) in a side–on view. The second configuration was used to quantify the fraction of secondary drops that froze after impact with the ice particle using two Raspberry Picameras (Raspberry PiCamera Module V2) referred to as RPicams, with a polarising filter (Standard 55 mm Circular Polariser) attached to one camera. At present, the two configurations are not compatible to work concurrently. Recordings using the high–speed camera were recorded at 1069 frames per second (fps) and recordings using the RPicams were recorded at 24 fps.

The experimental setup is operated in a cold room which can achieve a base temperature as low as -50 °C and provided the means of achieving a supercooled environment. The experimental setup was housed in a Bosch strut/Perspex panel frame to prevent the accidental introduction of frost particles during the experiments. A glass slide was supported on 3D printed plastic stilts approximately 10 cm in height which had a fan attached to dissipate the heat emitted from the polarised light source (LCD monitor). The temperature of the glass slide was monitored using a K–type thermocouple attached to the glass slide with aluminium tape. The relative humidity was not measured but will be below ice saturation, and possibly very small ice fragments were not observed due to sublimation preventing growth to visible sizes. The ice particles were prefabricated by freezing ultrapure water drops (Endotoxin–Free UP H2O, Merck) of approximately 6 mm in diameter on a glass slide coated in a water repellent (Rain–X) using a Peltier cooling system. The typical freezing shape of the ice particle is shown in Fig. 1. A pipette was modified to allow an ultrapure water drop (Endotoxin–Free UP H2O, Merck) at room temperature with a diameter of approximately 5 mm to be placed on the pipette using a disposable needle (22 gauge, sterile) and syringe. The modified pipette was held in a 3D printed tipper mechanism parallel to the glass slide, and the water drop was allowed to reach thermal

equilibrium with the cold room for 90 s. The supercooled drop was released from the modified pipette perpendicular to the glass slide and was controlled by an Arduino and servo motor.

As the drop height and initial supercooled water drop diameter before impact ($D$) were kept constant at 1.36 m and 5 mm respectively, the normal impact velocity ($V_0$) for all experiments was 5.2 m s$^{-1}$. The terminal velocity of a 5 mm diameter drop is approximately 9 m s$^{-1}$ (Gunn and Kinzer, 1949). Initially, the impact velocity may seem unrealistic. However, the ice particle in these experiments was held stationary on a glass slide, but in the atmosphere the ice particle would also be falling. The terminal velocity will depend on the ice particle shape, but for aggregates of similar size it is typically around 1 m s$^{-1}$ (Locatelli and Hobbs, 1974). Moreover, turbulence, especially in deep convective clouds, may also affect the impact velocity Pinsky and Khain (1998). While such large droplets are rare in the atmosphere the purpose here is to demonstrate that the process is a potential secondary ice mechanism. The supercooled water drop and the ice particle/glass slide were in thermal equilibrium for all experiments.

The temperature range investigated was between -4 °C to -12 °C. As the temperature of water decreases, the surface tension ($\sigma$) and viscosity ($\mu$) of water increases (Hrubý et al., 2014; Dehaoui et al., 2015). In fluid dynamics, the Weber number, We $= \rho D V_0^2 / \sigma$, and Reynolds number, Re $= \rho D V_0 / \mu$, are used to relate the inertial forces of the fluid to its interfacial and viscous forces respectively. In this case, the fluid is the supercooled water drop, and the diameter of the supercooled water drop, D, refers to the diameter before impact. The inertial force is from the initial impact velocity of the supercooled water drop, and the interfacial (surface tension) and viscous forces are properties of the supercooled water drop. Taking into account the temperature dependent values of surface tension and viscosity of the supercooled water between -4 °C to -12 °C, the We and Re number ranges obtained were $1747 \leq$ We $\leq 1772$ and $8781 \leq$ Re $\leq 12240$, respectively.

We conducted 32 experiments using the RPicams configuration during quantification of the freezing fraction of secondary drops and the data is given in Table A1.

## 3 Results

From our high–speed and RPicams recordings we present a schematic diagram of the formation of secondary drops from a supercooled water drop impact on an ice particle on a glass slide in Fig. 2. The We and Re numbers used were sufficiently large, i.e. We $\gg 2.5$ and Re $\gg 25$, such that inertia dominated the spreading of the thin film (Roisman, 2009). Surface tension and viscosity forces were therefore considered negligible during the spreading phase of the drop (Roisman, 2009), as was the wettability of the surface (Antonini et al., 2012). Figure 2(a) depicts the filament–like structures which were ejected during the spreading phase of the drop impact. We were unable to track the positions of these secondary drops or quantify them with our current high–speed camera or RPicams configurations. As the kinetic energy is transferred from that of a vertical to horizontal motion at impact, the water drop spread out radially, and instabilities at the rim were also observed. Figure 2(b) depicts the retraction of the drop, which caused the instabilities to 'pinch off' or rupture, followed by partial rebound. On superhydrophobic surfaces, rupturing of the instabilities has been attributed to surface tension (Zhang et al., 2020). Our glass

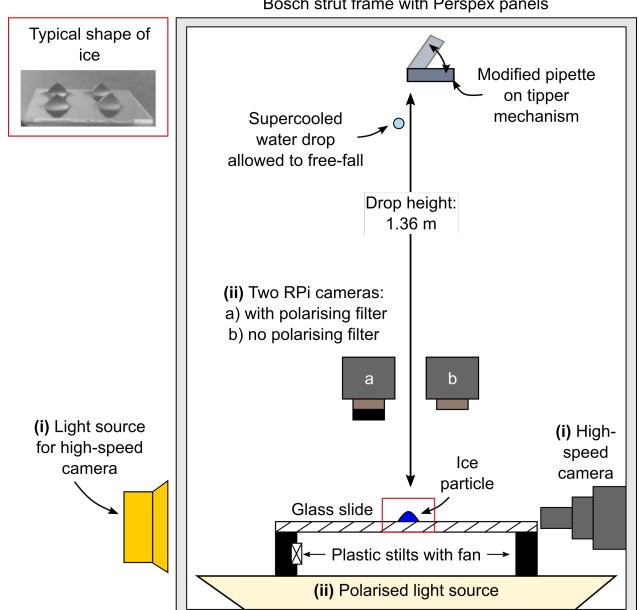

**Figure 1.** Schematic diagram of the experimental setup. Components labeled **(i)** were used in the high–speed configuration and **(ii)** were used in the RPicams configuration. The setup was operated in a cold room to achieve a supercooled environment.

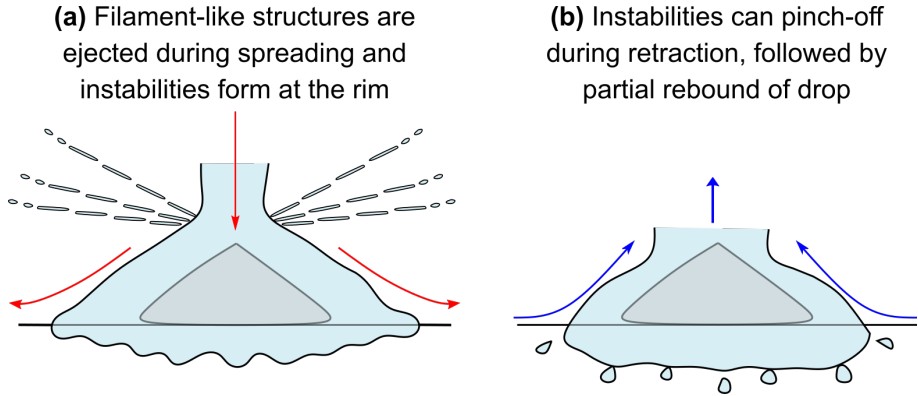

**(a)** Filament-like structures are ejected during spreading and instabilities form at the rim

**(b)** Instabilities can pinch-off during retraction, followed by partial rebound of drop

**Figure 2.** A schematic diagram of a supercooled water drop impact on an ice particle on a glass slide and subsequent secondary drop formation during (a) the spreading phase and (b) the retraction phase.

slide, coated in a water–repellent, is probably superhydrophobic, and surface tension is likely the cause of the rupture of the rim instabilities.

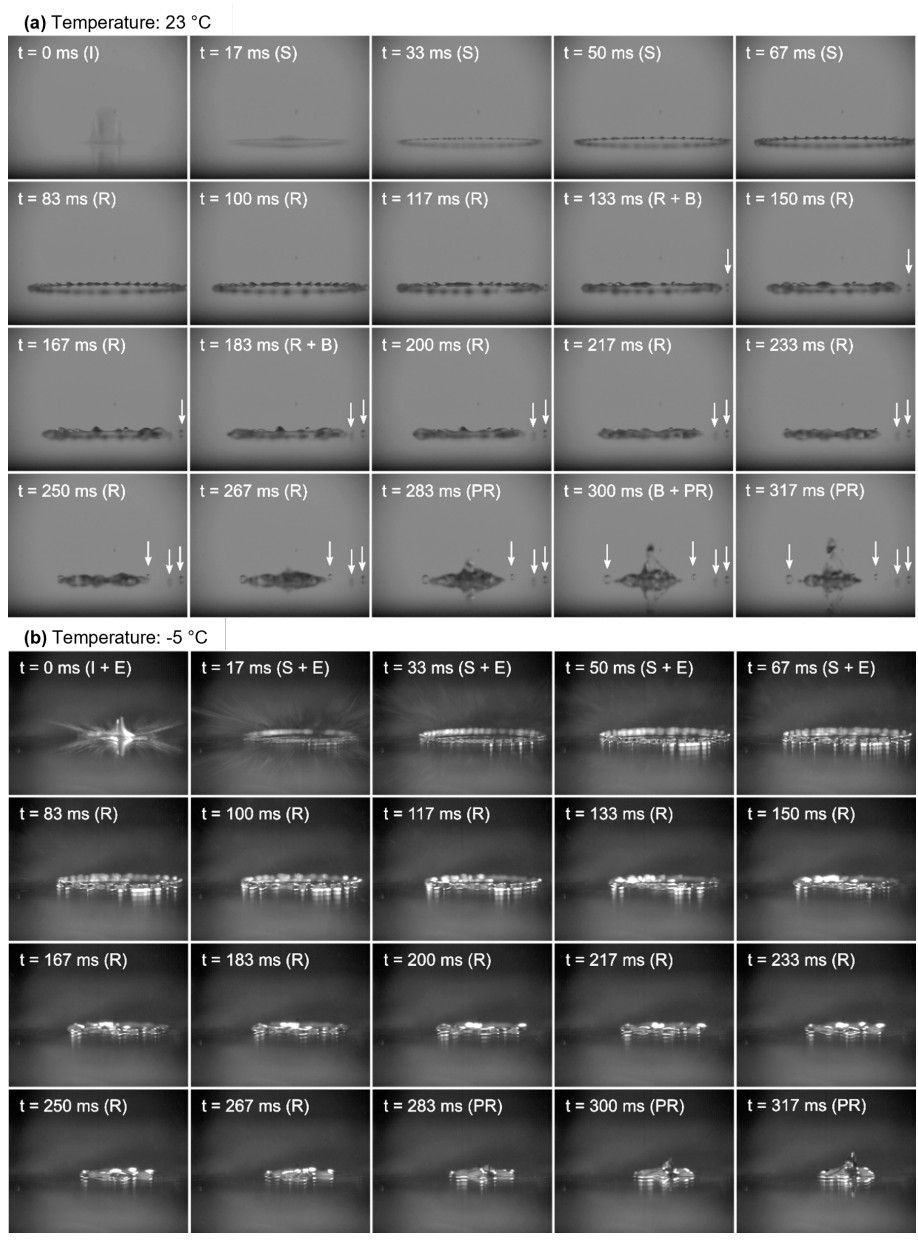

**Figure 3.** Frames from the high–speed camera configuration of a water drop impact on a glass slide when both water drop and glass slide are at (a) room temperature (23 °C) and (b) -5 °C. The impact phase (I), spreading phase (S), secondary drop formation/ejection during the spreading phase (E), retraction phase (R), secondary drop formation due to receding break-up (B) and partial rebound (PR) of the water drop are indicated in the frames. Arrows indicate secondary drop formation during the retraction phase of the water drop.

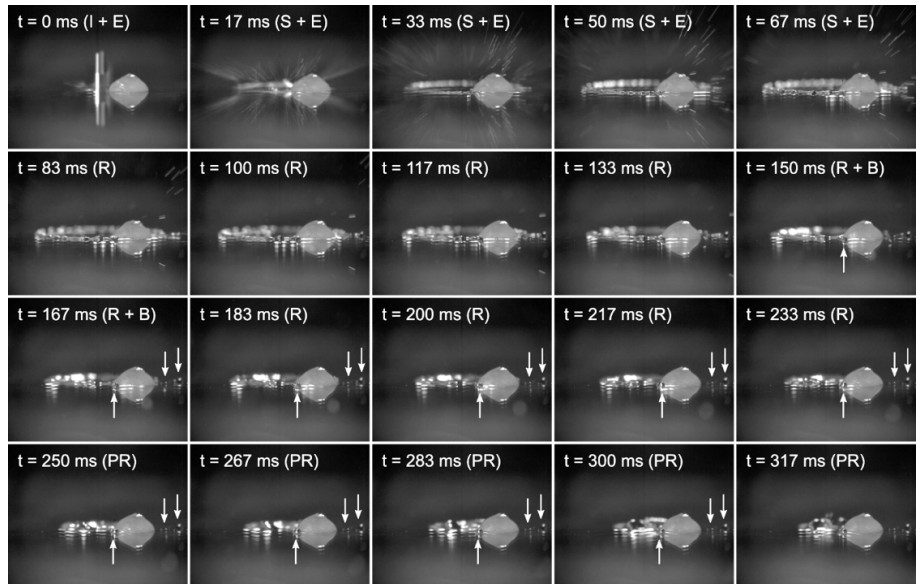

**Figure 4.** Frames from the high–speed camera configuration of a supercooled water drop impact on an ice particle when both drop and ice particle are at -5 °C. The impact phase (I), spreading phase (S), secondary drop formation/ejection during the spreading phase (E), retraction phase (R), secondary drop formation due to receding break-up (B) and partial rebound (PR) of the water drop are indicated in the frames. Arrows indicate secondary drop formation during the retraction phase of the supercooled water drop.

## 3.1 Drop impact: high-speed recordings

We performed control experiments at room temperature (23 °C) and several supercooled temperatures using the high–speed camera configuration to characterise the water drop (diameter of 5 mm) impacting the glass slide. Figure 3 shows the frames from a high–speed recording of (a) a water drop impact on the glass slide at room temperature and (b) a supercooled water drop impact at -5 °C.

On impact with the glass slide, the water drop deformed and spread radially outwards as a thin film bordered by a thicker rim. Instabilities at the rim were observed for both the room temperature drop and the supercooled drop at -5 °C. The supercooled drop shown in Fig. 3(b) ejected straight filament–like structures at an angle to the glass surface close to the impact and these filament–like structures disintegrated into secondary drops. This was in contrast to the impact of the water drop at room temperature where no ejection of filament–like structures was observed, perhaps due to higher viscosity and surface tension of water at supercooled temperatures. During the retraction phase, some of the rim instabilities pinched off from the thin film in the experiments with the water drop at room temperature forming secondary drops, in a process called 'receding break–up'. In contrast, no receding break–up was observed for the supercooled drop.

Figure 4 shows the frames of a supercooled water drop impacting the side of an ice particle at -5 °C. Similar to the super-cooled water drop on a glass slide, filament–like structures, which dissipated into secondary drops, formed at or close to impact with the glass slide/ice particle. Unlike the impact of a supercooled water drop on a bare glass slide, secondary drops formed

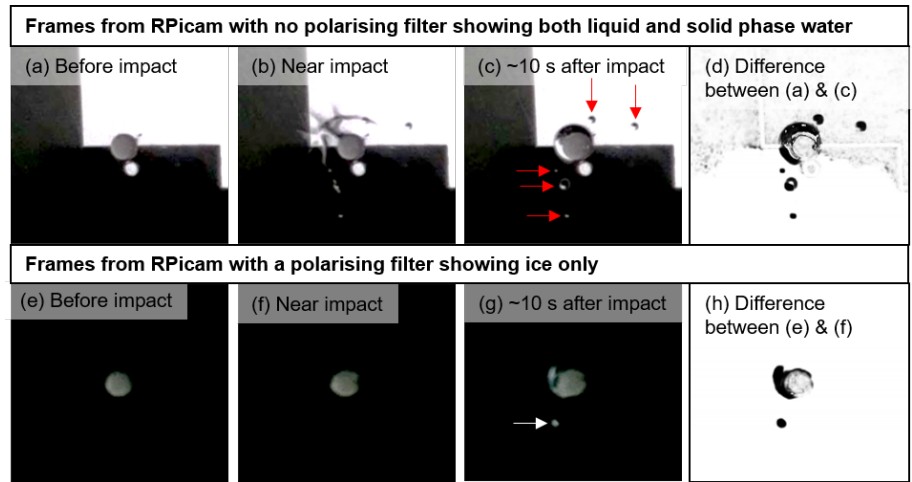

**Figure 5.** Selected frames from the impact of a supercooled water drop on an ice particle at -4°C using the RPicams configuration. Frames (a)–(c) before, at and ∼10 s after impact using the camera with no polarising filter. Red arrows in (c) indicate the number of secondary drops formed. Frame (d) shows the difference between (a) and (c). Frames (e)–(g) before, at and ∼10 s after impact using the camera with a polarising filter. The white arrow in (h) indicates the frozen secondary drop. Frame (h) shows the difference between (e) and (g).

via receding break–up. These secondary drops were observed around the parts of the rim of the thin film which contacted the ice particle.

## 3.2  Determining the freezing fraction of the secondary drops: RPicams

We performed supercooled water drop impacts on ice particles between -4 °C to -12 °C. To unambiguously identify if a
secondary drop had frozen, we used a polarising filter with a polarised light source, exploiting the birefringent properties of ice. Figure 5 shows selected frames of a supercooled water drop impact at -4 °C using the RPicams configuration. The top row of Fig. 5 shows frames from the camera with no polarising filter (a) before, (b) at and (c) ∼10 s after impact. The number of secondary drops observed are indicated by red arrows in Fig. 5 (c). The difference between the Fig. 5 (a) and (c) is presented in Fig. 5 (d) clearly indicating the secondary drops formed. The bottom row shows frames from the camera with a polarising
filter (e) before, (f) at and (g) ∼10 s after impact. The frozen secondary drop is indicated by a white arrow in Fig. 5 (g). The difference between the Fig. 5 (e) and (g) is presented in Fig. 5 (h) clearly indicating the frozen secondary drop formed.

For this particular experiment, five secondary drops formed, of which one froze, giving a freezing fraction, $\Phi = 0.2$. During these experiments, two types of supercooled water impacts occurred: direct impact on the ice particle and partial impact on the ice particle. These different impacts arose due to practical difficulties with consistently impacting the ice particle with
supercooled water drop due to changes in viscosity of water at different temperatures. For the experiment shown in Fig. 5, the impact was a side impact towards the top left of the ice particle as indicated in Fig. 5 (b). The RPicams configuration only observed the larger > 0.1 mm diameter drops formed during retraction of the thin film. The smaller secondary drops (< 0.1 mm

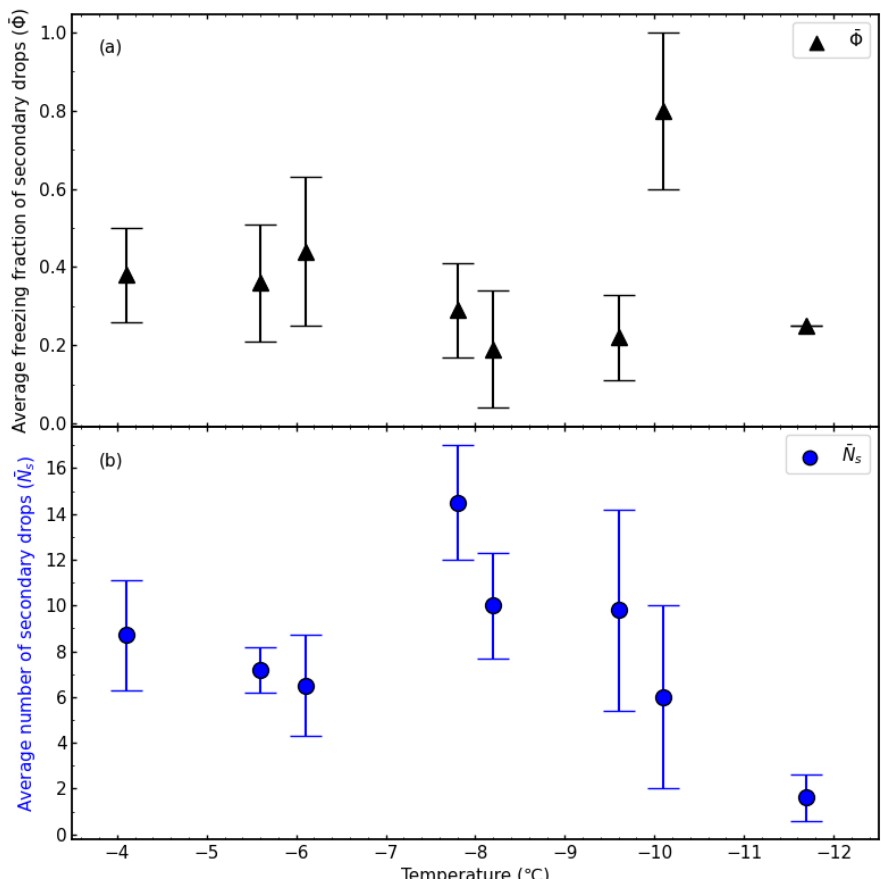

**Figure 6.** (a) The average freezing fraction of the secondary drops, $\bar{\Phi}$ (black triangles) and (b) the average number of secondary drops, $\bar{N}_s$ (blue circles) as a function of temperature. Average data included both direct and partial collisions. The error bars represent the standard error in the freezing fraction or secondary drop number for the temperature intervals which are listed in Table A2 & A3.

diameter) observed at impact from the high–speed configuration were not observed using this configuration as the minimum drop diameter the RPicams could detect was 0.1 mm.

160  Figure 6(a) shows the average freezing fraction of secondary drops formed when a supercooled water drop with a diameter of 5 mm collided with an ice particle, $\bar{\Phi}$, as a function of temperature. The raw data can be found in Table A1 and the averaged data of the freezing fraction of secondary drops in Table A2. The average number of secondary drops, $\bar{N}_s$, liquid or solid, shown in Fig. 6(b) as a function of temperature, which reached a maximum at approximately -7.5 °C. The averaged data of the number of secondary drops can be found in Table A3.

## 4  Discussion

We discuss some aspects of the experimental setup that may affect the occurrence and rate of secondary drop production and freezing here.

As the ice particles were placed on a flat glass slide, during impact, the supercooled water drop spread across the ice particle and on to the glass slide where the larger > 0.1 mm diameter sized secondary drops formed. We acknowledge that the glass slide presents an artificially flat surface compared to atmospheric conditions. However, a study by Schremb et al. (2018) showed that, on an elevated ice surface, the thin film of a supercooled water drop with a diameter of ∼4 mm and similar We and Re numbers at -14 °C was ejected and subsequently ruptured, forming secondary drops. While quantification was not the focus of their study, it was observed that the rim of the supercooled water drop was largely frozen, and only some of the secondary drops were observed as ice. The size of the secondary drops formed in the study by Schremb et al. (2018) is comparable to our secondary drops despite the different generation mechanism, and the supplementary videos indicate that 10s of secondary drops were formed. Furthermore, water drops with diameters between ∼3–4 mm colliding with a steel disk of ∼4 mm in diameter (Rozhkov et al., 2002) and water drops with diameters of 6 mm colliding with an iron cylinder of the same diameter (Villermaux and Bossa, 2011) produced 100s of secondary drops. Clearly, secondary drops still form, emitted from the rim of the thin film during impact, when there is no supporting flat surface, such as the glass slide used in this study. However, there is much uncertainty about the number of secondary drops formed.

In addition, the ice particle in our experiments is in a fixed position on the glass slide, whereas, in the atmosphere, the ice particle is in free fall. When the faster-moving supercooled water drop collides with the ice particle, the ice particle will move in response to the collision, likely affecting the formation of the secondary drops and their subsequent freezing. However, currently, it is difficult to ascertain how this will influence secondary drop formation and freezing without further investigations into the mechanisms of secondary drop formation on an elevated ice particle.

Another factor that will influence the generation of secondary drops is the ice particle shape. Our ice particles have a pointed tip, as shown in Fig. 2, which is a typical shape formed when a liquid water drop is frozen on a cold substrate (Snoeijer and Brunet, 2012), but not representative of atmospheric ice particles. According to Phillips et al. (2018), who refer to this SIP mechanism as 'Mode 2', for it to occur, the supercooled water drops must have a diameter larger than 150 μm and the ice particle more massive still. In the atmosphere, ice particles which are larger than 150 μm are typically irregular in shape (Korolev and Sussman, 2000). A study by Zhang et al. (2020) shows that at room temperature, water drop impact on curved surfaces induce additional fragmentation mechanisms compared to flat surfaces. Therefore, we expect the irregular shape of an ice particle to introduce additional fragmentation mechanisms of the supercooled water drop which may enhance secondary drop formation.

We observed a decrease in the number of secondary drops formed during receding break–up as temperature decreased below -8 °C. Figure 7 shows the frames after a supercooled water drop impact with an ice particle for the experiments between -11°C to -12°C which was the range where the smallest number of secondary drops formed. At these temperatures, the supercooled water drop froze either during the spreading phase or in the early stages of the retraction phase. As the growth velocity of ice in

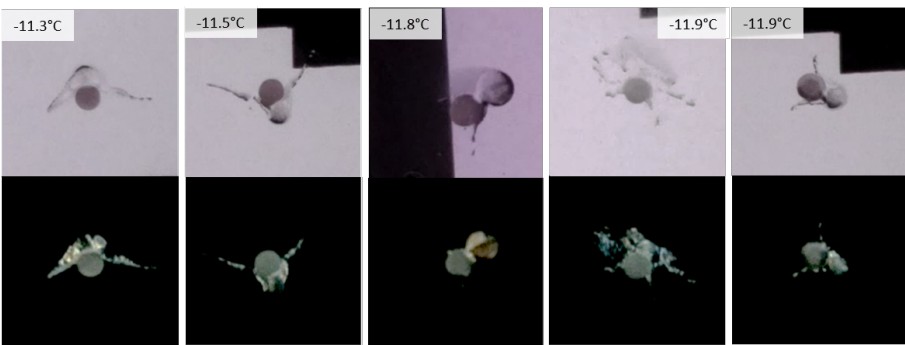

**Figure 7.** Frames from the RPicams configuration approximately 10 s after a supercooled water drop impact for experiments at T ≤ -11 °C. The top panel shows frames from the RPicam with no polarising filter and the bottom panel shows frames from the RPicam with a polarising filter.

supercooled water increases with decreasing temperature, e.g. at -2 °C it is around 0.2 cm s$^{-1}$, whereas at -10 °C it is around 5 cm s$^{-1}$ (see Pruppacher and Klett, 1997, chapter 16), which may explain why a decrease in secondary drops was observed. We believe the decrease in secondary drop formation at temperatures below -8 °C may be due to the artificially flat geometry presented by the glass slide and to the large size of the incident drop, both factors which prolonged the interaction time between the supercooled water drop and ice. For example, the supplementary videos from Schremb et al. (2018) showed 10s of secondary drops forming at -14 °C after impact on an elevated ice target, more than we observed at our lowest temperature of -12 °C.

Whilst the freezing mechanism of the secondary drops was not specifically studied in this work, we consider the following mechanisms. The freezing of supercooled water drops occurs in two stages. The first stage is characterised by the formation of ice dendrites throughout the supercooled water drop. The latent heat from the formation of the ice dendrites is released during this stage, warming the temperature of the supercooled water drop to ~0 °C. The second stage is characterised by the freezing of the remaining supercooled water drop and is controlled by the loss of latent heat due to the supercooled water drop surroundings. Stage 1 of freezing is fast and the time taken for this stage to complete ($t_i$) can be estimated from the following equation (Macklin and Payne, 1967):

$$t_i \approx \frac{\delta_R}{G} \tag{1}$$

where $\delta_R$ is the thickness of the layer of supercooled water on the ice particle and $G$ is the growth velocity of ice which is temperature dependent.

From Fig. 4, we can estimate that the rim of the supercooled water drop, which is also the thickest part of the supercooled water drop, is approximately 0.78 mm. Taking this value for $\delta_R$ and given that the growth velocity of ice at -5 °C is approximately 1 cm s$^{-1}$ (Pruppacher and Klett, 1997, chapter 16), then $t_i \approx 0.078$ s. Figure 4 shows the time-scale for the retraction phase is of the order 0.1 s. It is plausible that the initial ice dendrites can propagate through the supercooled water drop and that water containing these ice dendrites may then break off during the retraction phase and initiate freezing. The second phase

of freezing will take longer, but as long as the drop contains ice dendrites it will eventually freeze. This explanation is also proposed by Schremb et al. (2018) and Phillips et al. (2018) who suggest that seeding ice crystals are transported during the initial spreading phase. Alternative freezing mechanisms include the formation of a thin, unobserved film of liquid water present on the glass slide after the retraction phase. The contact between the thin film of water and the ice particle could induce freezing in the thin film, which could then trigger freezing in the seemingly detached secondary drop. Mechanical agitation or shock may also play a role in the freezing of the secondary drops (Alkezweeny, 1969; Czys, 1989). Regardless of the freezing mechanism, the glass slide will likely have some influence, and it will be pertinent to remove this in future investigations.

As a proof-of-concept investigation, we studied supercooled water drops with diameters of 5 mm and ice particles with diameters of 6 mm as larger sizes of supercooled water drops were easier to work with experimentally. While these sizes are not necessarily representative of cloud conditions, theoretically, this new SIP mechanism should occur where supercooled water drop diameters are > 150 μm and the ice particles more massive still. Supercooled water drops and ice particles are present within a variety of different clouds. For example, Hobbs and Rangno (1990) presented aircraft observations in small polar–maritime cumuli that displayed ice enhancement with cloud bases too cold for large cloud-droplets (> 24 μm) between -3 and -8 °C as required for rime splintering. Their discussion highlighted that ice enhancement proceeded in two stages. The first stage consisted of the formation of frozen drops, < 400 μm diameter, and small graupel particles, < 1 mm diameter. The second stage was characterised by the appearance of high concentrations of vapour–grown ice crystals in the upper regions of the cloud. A key finding of this series of papers was that high concentrations of small ice particles appeared simultaneously with frozen drizzle drops. Furthermore, Rangno and Hobbs (2001) showed that large supercooled drops were often a requirement for ice enhancement in moderately cooled Arctic stratiform clouds, and ice enhancement was often coincident with observations of large supercooled raindrops. Supercooled drizzle drops and raindrops are common in convective clouds (e.g. Crawford et al., 2012; Taylor et al., 2016), as are large ice particles. Hence, because there is a broad continuum of drizzle and raindrop sizes, where the larger drops freeze first, followed by accretion of the smaller unfrozen drops indicates that collisions of supercooled water drops with ice particles more massive may be of importance in a wide range of clouds.

## 5    Conclusions and Future Work

In this study, we confirmed that during collisions of supercooled water drops with ice particles, frozen secondary drops formed over the temperature range between -4 °C to -12 °C. Our main findings are:

1. Approximately 5 to 10 secondary drops are formed during the receding break-up of the retraction phase of a supercooled water drop (D = 5 mm) after collision with an ice particle (D = 6 mm) placed on a glass slide.

2. An average of 30% of these secondary drops formed froze between -4 °C to -12 °C.

3. Experiments with a high–speed camera highlighted that secondary drops formed as a jet of smaller droplets produced separately from the receding break–up of the drop. No quantification of the freezing fraction of the secondary drops can currently be made.

One of the main experimental challenges of this work was dropping the supercooled water drop consistently onto the ice particle which limited the amount of experiments we could perform. As shown in Table A1, the majority of the successful impacts were classified as partial hits despite the intention for them to be direct hits. While partial hits are expected in clouds, as well as direct hits, we also conducted many experiments where the supercooled water drop missed the ice particle. One method of achieving better control of the supercooled water drop impact could be via growth and supercooling of a water drop at the end of a needle similar to the system shown in Schremb et al. (2018). Compared to our current mechanism, which involved tilting a pipette to allow the supercooled water drop to roll off, the supercooled water drop would remain fixed to a certain point before detaching under gravity, making it easier to drop consistently in the same position.

Another experimental challenge we would like to address is quantifying the secondary drops formed during the spreading phase of the supercooled water drop during impact. Thoroddsen et al. (2012) quantified secondary drops ejected with velocities of up to 100 m s$^{-1}$ using an ultra-high-speed camera capable of recording at 1000000 fps, and we could use a similar setup. We could then exploit the birefringent properties of ice to determine whether these ejected secondary drops froze.

The number of secondary drops per collision is sensitive to geometry and material of collision, even for drops of the same size. We quantify about 10 secondary drops per collision. Schremb et al. (2018) observed on the order of tens of drops per collision for impacts on elevated ice surface. Finally, Rozhkov et al. (2002) observe 100s for drop impacts on steel disks at room temperature, as do Villermaux and Bossa (2011) for drop impacts on iron cylinders at room temperatures. Consequently, after addressing the above challenges and elevating the ice particle off the glass surface, which may be achieved simply by fixing the ice particle on a wire, further work is needed to investigate, more systematically, this new SIP mechanism over a range of experimental parameters, not limited to: supercooled drop sizes, supercooled water drop-to-ice particle size ratios, ice particle shapes, temperatures, drop height (and hence impact velocity), airflow, relative humidity conditions and chemical compositions of the supercooled water drop.

# Appendix A

**Table A1.** Total number of secondary drops and the number of frozen secondary drops for each experiment

|         | Temperature (°C) | Total number of secondary drops | Frozen secondary drops |
|---------|------------------|---------------------------------|------------------------|
| Direct  | -4.2             | 14                              | 1                      |
|         | -4.2             | 0                               | 0                      |
|         | -5.3             | 7                               | 6                      |
|         | -5.5             | 10                              | 2                      |
|         | -7.8             | 12                              | 2                      |
|         | -9.9             | 7                               | 0                      |
| Partial | -3.8             | 16                              | 5                      |
|         | -4.0             | 5                               | 1                      |
|         | -4.0             | 8                               | 5                      |
|         | -4.3             | 9                               | 6                      |
|         | -5.6             | 5                               | 0                      |
|         | -5.6             | 5                               | 1                      |
|         | -5.8             | 9                               | 5                      |
|         | -6.0             | 4                               | 1                      |
|         | -6.0             | 8                               | 2                      |
|         | -6.1             | 2                               | 1                      |
|         | -6.1             | 12                              | 3                      |
|         | -7.7             | 17                              | 7                      |
|         | -8.0             | 5                               | 0                      |
|         | -8.0             | 11                              | 7                      |
|         | -8.1             | 8                               | 1                      |
|         | -8.5             | 16                              | 0                      |
|         | -9.4             | 0                               | 0                      |
|         | -9.4             | 21                              | 6                      |
|         | -9.8             | 11                              | 4                      |
|         | -10.0            | 2                               | 2                      |
|         | -10.1            | 10                              | 6                      |
|         | -11.3            | 0                               | 0                      |
|         | -11.5            | 4                               | 1                      |
|         | -11.8            | 4                               | 1                      |
|         | -11.9            | 0                               | 0                      |
|         | -11.9            | 0                               | 0                      |

**Table A2.** The mean ($\bar{\Phi}$) and standard deviation ($\sigma$) of the fraction of frozen secondary drops within a specified temperature interval (T interval) along with the number of experiments (n) within the T interval, the average degree of supercooling within the temperature interval ($\bar{T}$) and the error in the sample mean ($\sigma_{\bar{\Phi}}$).

| T interval (°C) | | $\bar{T}$ (°C) | $\bar{\Phi}$ | $\sigma$ | n | $\sigma_{\bar{\Phi}}$ |
|---|---|---|---|---|---|---|
| -3.8 | -4.3 | -4.1 | 0.38 | 0.26 | 5 | 0.12 |
| -5.3 | -5.8 | -5.6 | 0.36 | 0.34 | 5 | 0.15 |
| -6.0 | -6.1 | -6.1 | 0.44 | 0.38 | 4 | 0.19 |
| -7.7 | -7.8 | -7.8 | 0.29 | 0.17 | 2 | 0.12 |
| -8.0 | -8.5 | -8.2 | 0.19 | 0.30 | 4 | 0.15 |
| -9.4 | -9.9 | -9.7 | 0.22 | 0.19 | 3 | 0.11 |
| -10.0 | -10.1 | -10.1 | 0.80 | 0.28 | 2 | 0.20 |
| -11.3 | -11.9 | -11.7 | 0.25 | 0.00 | 2 | 0.00 |

**Table A3.** The mean ($\bar{N}_s$) and standard deviation ($\sigma$) of the number of secondary drops within a specified temperature interval (T interval) along with the number of experiments (n) within the T interval, the average degree of supercooling within the temperature interval ($\bar{T}$) and the error in the sample mean ($\sigma_{\bar{N}_s}$).

| T interval (°C) | | $\bar{T}$ (°C) | $\bar{N}_s$ | $\sigma$ | n | $\sigma_{\bar{N}_s}$ |
|---|---|---|---|---|---|---|
| -3.8 | -4.3 | -4.1 | 8.7 | 5.9 | 6 | 2.4 |
| -5.3 | -5.8 | -5.6 | 7.2 | 2.3 | 5 | 1.0 |
| -6.0 | -6.1 | -6.1 | 6.5 | 4.4 | 4 | 2.2 |
| -7.7 | -7.8 | -7.8 | 14.5 | 3.5 | 2 | 2.5 |
| -8.0 | -8.5 | -8.2 | 10.0 | 4.7 | 4 | 2.3 |
| -9.4 | -9.9 | -9.6 | 9.8 | 8.8 | 4 | 4.4 |
| -10.0 | -10.1 | -10.1 | 6.0 | 5.7 | 2 | 4.0 |
| -11.3 | -11.9 | -11.7 | 1.6 | 2.2 | 5 | 1.0 |

*Acknowledgements.* We acknowledge the US Department of Energy (DE-SC0018932)for funding the main part of this study. We also acknowledge funding from NERC (NE/T001496/1).

*Data availability.* All datasets are provided in Appendix A.

*Video supplement.* All video recordings from the high–speed configuration and RPicams configuration are deposited in Figshare, a FAIR-aligned data repository and can be accessed at https://doi.org/10.48420/c.5476557.

*Author contributions.* VTJP and PJC conceived the original study. RLJ and PJC designed the new experimental setup with advice from VTJP. RLJ and PJC performed the experiments. RLJ analysed the data and wrote the paper. VTJP and PJC provided comments on the paper.

*Competing interests.* The authors declare that they have no conflict of interest

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
