# Peer review of "Secondary ice production during the break-up of freezing water drops on impact with ice particles"

_Atmospheric Chemistry and Physics, 2021_

## Referee Comment (RC2)

I enjoyed reading this paper, which presents significant new experimental results relating to secondary ice processes, and is certainly worth publishing. I have a few minor questions and suggestions for making the paper a bit stronger (see below).

*Active between -3 °C ≤ T ≤ -8 °C, rime–splintering occurs when supercooled water drop diameters are < 13 µm or > 24µm (Hallett and Mossop, 1974; Mossop and Hallett, 1974; Mossop, 1978)*

Maybe I misremember the Mossop 1978 paper, but should the condition on droplet sizes here be "< 13 µm **and** > 24µm" rather than **or** ?

Section 2

Perhaps it could be useful just to elaborate a little bit on these formulae. Maybe showing a figure with graphs of Ns vs DE; Phi vs T; f(T) would help make the general behaviour clearer?

*The typical freezing shape of the ice particle is shown in Fig. 1.*

Is the shape of the ice particle likely to be a relevant factor here? How could you find out? What is it likely to be in the atmosphere?

*the impact velocity (V0) for all experiments was 5.2m s$^{-1}$*

it's worth pointing out in the text that this is below terminal velocity for a 5mm drop (which would be closer to 9 m/s). However in the real atmosphere the ice particle would be moving as well, so the differential velocity may be more realistic than it might initially appear.

*In fluid dynamics, the Weber number, $We = \rho D V_0^2/\sigma$, and Reynolds number, $Re = \rho D V_0/\mu$, are used to relate inertial forces to interfacial and viscous forces, respectively. Taking into account the temperature dependent values of surface tension and viscosity of the supercooled water between -4 °C ≤ T ≤ -12 °C, the We and Re number ranges obtained were 1747 ≤ We ≤ 1772 and 8781 ≤ Re ≤ 12240, respectively.*

I think in both cases here, it would be good to clarify what We and Re refer to – or more specifically *where* these inertial, viscous, and interfacial forces are acting. Often in cloud physics we think about the inertial, viscous in the air surrounding the drop, while here (I think) you are considering them *within* the water

Is it obvious what the length scale and velocity scale in We and Re should be? You have chosen $V_0$ for the velocity scale, so that implies the water fluid parcels of interest are moving at this velocity. So are you considering the downward motion of the liquid water at the moment of impact on the ice particle? Or the lateral velocity of the liquid water as it spreads out? (are these velocity scales comparable?).

For the length scale, it's not obvious what to choose, when you have a liquid spreading over a solid surface. The depth of the water coating? D is probably not an unreasonable choice, but maybe you can make the argument a bit more explicit somehow. Again, it all comes down to what aspect of the flow of the water you are trying to characterise.

Section 3 – you used a high speed camera. What exposure time was used? It seems from the images like the splash itself (t=0) is quite blurred. Was this limited by the illumination?

For figs 2,3,4 I did wonder whether adding some slightly more detailed description of what's happening in the various frames would help the reader interpret what they are seeing. It took me a while to get a sense of what was happening. Or maybe some extra annotation on the figures themselves?

Discussion - You mention the influence of the glass slide, and I agree the presence of the slide itself is definitely worth discussing. Another factor I can think of here is that the ice particle is effectively in a fixed vertical position, while in the atmosphere the ice particle is in free fall, and when the drop hits it, then the ice particle can move in response to that – so some of the drop's momentum can be carried to the ice particle. Would that change the way the water flows over the ice particle, and freezes?

In figure 5 I think it's important to clarify what the error bars represent in the caption, and in the text. Is it the variation from one experiment to the next, in the "same" conditions? Or is it the uncertainty on the mean value?

Connected to this is Table A2 – the values of phi, sigma, and sigma_phi_bar are all quoted to the nearest 0.1, which seems a bit coarse. Might be worth 1 extra significant figure?

The number of experiments is fairly small, given the variability in phi that's shown. I'm guessing these are quite time consuming to conduct and analyse. Perhaps you can discuss that a bit? In general I would enjoy seeing an expansion of the future work in section 6 to talk about how the experiment could be improved and elaborated. Likewise saying "*no quantification of the freezing fraction of the secondary ice drops [from the jet of smaller droplets] can currently be made*" is fine, but it would be good to discuss what you would need to do to quantify it, or study it in more detail.

---

## Referee Comment (RC3)

**Review of** *Secondary ice production during the break-up of freezing water drops on impact with ice particles*
Sylvia Sullivan

James, Phillips, and Connolly present experimental results on a secondary ice production (SIP) process involving liquid drop-ice crystal collisions. It is nice to see additional, and especially quantitative, laboratory results on SIP, and I support publication of the article. I feel, however, that several points should be elaborated and that some reorganization of sections would help with clarity.

**Major Comments**
- Given that a condensed version of the theoretical work in Phillips et al. 2018 was presented, I expected there to be some comparison of the observational results with this model. For example, can you calculate Ns as a function of temperature from the Phillips et al. model and overlay it on Figure 5? Can you say anything about the validity / assumptions of the model on the basis of these experiments?

- I felt that the mechanistic discussion at the start of Section 5 would have been helpful prior to Sections 4.1 and 4.2, so that I had a better sense of what was physically happening in the experiments in Figs 2 and 3.

- I appreciate that the limitation of the glass slide is acknowledged and secondary drop production in other setups is discussed (lines 159-171). But does the presence of the glass slide also mean that the mechanism shown in Figure 6 may need to be modified in the real atmosphere? In particular, given that the secondary drops and frozen fraction have only been quantified in the retraction phase, should an equivalent retraction phase exist between two curved surfaces in relative motion?

- Section 5.1 read like introductory material, as it did not include discussion of any of the study's findings. It is valuable information, but I would integrate it into the introduction as general motivation to study SIP.

- In place of Section 5.1, I think a new (small) "Atmospheric Implications" section or additional sentences throughout the Discussion / Conclusion would be helpful to discuss the atmospheric conditions under which the experiments are representative, i.e. In what regions / synoptic conditions / cloud systems, would ice particles of diameter 6 mm and raindrop of diameter 5 mm coexist? For what range of ice particle and raindrop sizes, does 5.2 m s-1 represent a realistic relative terminal velocity? Would it be possible to use the We and Re characterizing these experiments to identify regimes in in-situ data for which this mode 2 fragmentation could occur? etc.

**Minor Comments**
Line 15 – "where *subzero* temperatures"

Lines 17-18 – "typically *fall between* 1 x 10-5 L-1 and 1 L-1 at temperatures T~-10 deg C" (looking at Fig. 1-10 from Kanji et al. 2017)

Line 24 – "NWP models underestimate the concentrations of ice particles" – It would be nice to include an order-of-magnitude range for these underestimates.

Line 28 – "supercooled water drop diameters are < 13 um *and* > 24 um" In Hallett and Mossop 1974, both droplet sizes should coexist.

Line 31 – Along with the temperature range for frozen droplet shattering, it would be worthwhile to include a droplet size range as well, since droplet size will be discussed later as a parameter of the current experiments (e.g. 280-350 um in Keinert et al. 2020)

Line 34 – I would suggest to rephrase as "the attention *of laboratory studies has overwhelmingly* focused on the rime-splintering…", since a growing body of recent work has look at breakup parameterizations (e.g. Hoarau et al. 2018, Sotiropoulou et al. 2020, Sotiropoulou et al. 2021, Dedekind et al. 2021, etc.)

Line 52 – If you note that 'Mode 2' of frozen droplet fragmentation is studied, it would be helpful to know what 'Mode 1' is also.

Line 64 - "D$E_{crit}$" (not D$_{crit}$). Also you have not yet defined the freezing stages when you mention "stage 1 of freezing" here and again in Line 68.

Line 70 – "Finally, Phillips et al. 2018 hypothesised that $\Phi(T) = \min[4f(t),1]$ such that $\Phi = 0.5$ at -10 C" Stated like this, it sounds rather ad hoc. Perhaps an additional sentence can clarify where this form comes from or how it is constrained.

Line 94 – I do not know how important it is, but it was not clear to me what the "x-y translator (modified 3D printer)" was in the setup.

Line 99 – Surface tension of the liquid water is presented as $\gamma$ in line 61 and $\sigma$ here; viscosity is presented as $\lambda$ here and $\mu$ in the equation for the Reynolds number.

Lines 113-114 – It was not clear to me why the filament-like structures do not form when the colliding droplet spreads on the glass slide at room temperature. Is there a physical explanation for why this only occurs at colder temperatures?

Section 4.1 – Somewhere in this section or perhaps in the preceding Section 3, it would be helpful to have already referred to Table A1, so that it is clear from how many experiments the results come, e.g. only two glass slide collisions total were performed at 23 and -5 C?

Lines 133-134 – Was anything learned from the partial versus direct collisions? Does one or the other produce more secondary drops or higher frozen fraction? I guess there may be no robust difference, given the difficulty of performing these direct collisions. Also was Figure 5 produced from all data (both partial and direct) in Table A1? This should be specified in the caption.

Lines 136-138 – "The smaller secondary drops observed at impact … were not observed." This seems like it may be an important limitation. Is there the possibility to improve RPicam resolution in future work? This should be mentioned in the conclusions / future work if so.

Line 148 – "Surface tension and viscosity forces were considered negligible during the spreading phase of the drop" I am confused by this statement. Where / in which calculations are these forces being considered negligible?

Lines 150-151 – I have not seen the prompt-type / corona-type splash terminology before; I would define these terms more completely from the citations in these lines.

Figure 7 – Are the top versus bottom panels also with and without the polarising filter?

Line 176 – "T less than equal to *-11 deg C*"

Lines 180-187 – I find the arguments here difficult to follow. The takeaway is that temperature dependence of frozen fraction is caused by a liquid-ice interaction time scale? Could the authors reword somehow for clarity?

Line 189 – "We believe that the freezing fraction of the secondary drops is independent of the number of drops formed." Is there a reason for this belief? I would expect temperature dependence to dominate also, but I could also imagine that when a fixed fraction of the colliding droplet mass produces secondary drops, and more such secondary drops form, they are smaller and freeze faster..?

**Citations**

Dedekind et al. 2021Sensitivity of precipitation formation to secondary ice production in winter orographic mixed-phase clouds. Atmos. Chem. Phys. Disc.

Hoarau et al. 2018 A representation of the collisional ice break-up process in the two-moment microphysics LIMA v1.0 scheme of Meso-NH. Geosci. Model Dev. **11** 4269-4289

Kanji et al. 2017 – Cited in manuscript

Keinert et al. 2020 Secondary ice production upon freezing of freely falling drizzle droplets. J. Atm. Sci. **77**(8) 2959-2967

Korolev and Leisner 2020 – Cited in manuscript

Sotiropoulou et al. 2020 The impact of secondary ice production on Arctic stratocumulus. Atmos. Chem. Phys. **20** 1301-1316

Sotiropoulou et al. 2021 Ice multiplication from ice-ice collisions in the high Arctic: sensitivity to ice habit, rimed fraction, ice type, and uncertainties in the numerical description of the process. Atmos. Chem. Phys. **21** 9741-9760

---

## Author Comment (AC1)

**AC Response to RC1**

We thank the referee for their time in reviewing our manuscript and appreciate the constructive feedback given.

**RC:** Secondary ice production is an important topic, so I support publication of this paper. The authors have shown, convincingly, that collision of a supercooled droplet of water with a larger ice particle can produce secondary droplets. The authors have shown, less convincingly in my opinion, that those secondary droplets might freeze.

In essence, the experiments described in this paper are a refinement of those described in the JFM paper by Schremb et al. (the authors cite this paper). The authors here have documented some aspects of liquid water-ice collisions that Schremb et al. did not, and have quantified some others. My primary concern in the interpretation of these results is the freezing of the secondary drops. I am convinced, both from this paper and from Schremb's, that secondary drops are produced from instabilities in the rim of the water drop as it splashes across the ice. The filament structures shown in Fig. 6 are also a potential source of secondary droplets, though the number produced wasn't/couldn't be quantified in this study.

It wasn't clear to me from the manuscript whether the secondary drops froze in the air, or whether they froze once they landed on the substrate. If they froze once on the substrate, it is highly likely that freezing was because they are on the substrate. The probability that droplets of that size will freeze at temperatures in the range of approximately -10 C is very low. (This is true if they are on the substrate, but especially true if they are not.) Because the original ice is sitting on the substrate, there's the chance that a very thin film of ice can propagate along the surface from the ice to the supercooled liquid of the secondary droplet and cause freezing. (The ice that propagated along the substrate might not be apparent.)

**AC**: We believe that the drops we observe on the substrate are only formed during the retraction phase of the supercooled water drop and froze on the substrate. We did test drop impacts on the substrate without the presence of the ice particle and did not observe freezing within 30 min, whereas, for drop impact onto the ice particle freezing was observed within 10 s after impact. Therefore, this suggests that the substrate does not cause supercooled drops to freeze.

**RC:** The authors do address one possibility of how those smaller, secondary drops might freeze at temperatures as high as their experiments – shear at the ice-liquid interface which breaks off an ice embryo which causes freezing in the secondary droplet. I find this explanation unconvincing. (Something like this is also alluded to in Schremb's paper. I find it unconvincing there too.) If we impose a no-slip boundary condition at the solid-liquid interface, which we usually do, there's no shear at the interface. The shear is all in the liquid. If there's a frost-like layer on the ice, pieces of that might break off into the liquid that becomes a secondary droplet, I suppose. If that were to the case, I would expect freezing of the front, not necessarily freezing of the secondary droplet once it detaches. If the freezing mechanism is in fact shearing of an embryo into the secondary droplet, you can estimate some typical time scales. You know the time scale for detachment from the measurements. See section 16.1.4 in Pruppacher and Klett for some thoughts on freezing time for droplets.

**AC:** We thank the referee for their insight. We have removed the paragraph where we suggest that freezing could be due to shear and have added the following paragraph:

'Whilst the freezing mechanisms of the secondary drops was not specifically studied in this work, we consider the following mechanism. The freezing of supercooled water drop occurs in two stages. The first stage is characterised by the formation of ice dendrites throughout the supercooled water drop.

The latent heat from the formation of the ice dendrites is released during this stage, warming the temperature of the supercooled water drop to ~0 °C. The second stage is characterised by the freezing of the remaining supercooled water drop and is controlled by the loss of latent heat due to the supercooled water drop surroundings. Stage 1 of freezing is fast and the time taken for this stage to complete (ti) can be estimated from the following equation (Macklin and Payne, 1967):

$$t_i \approx \frac{G}{\delta R}$$

where $\delta R$ is the thickness of the layer of supercooled water on the ice particle and $G$ is the growth velocity of ice which is temperature dependent.

From Fig. 4, we can estimate that the rim of the supercooled water drop, which is also the thickest part of the supercooled water drop, is approximately 0.78~mm. Taking this value for $\delta R$ and given that the growth velocity of ice at -5 °C is approximately 1 cm s$^{-1}$ (Pruppacherand and Klett, 1997, chapter 16) then t$_i$ = 0.078 s. Figure 4 shows the time-scale for the retraction phase is of the order 0.1 s. It is plausible that the initial ice dendrites can propagate through the supercooled water drop and that water containing these dendrites may then break off during the retraction phase and initiate freezing. The second phase of freezing will take longer, but as long as the drop contains ice dendrites it will eventually freeze. This explanation is also proposed by Schremb et al. (2018) and Phillips et al. (2018) who suggest that seeding ice crystals are transported during the initial spreading phase. Alternative freezing mechanisms include the formation of a thin, unobserved film of liquid water present on the glass slide after the retraction phase. The contact between the thin film of water and the ice particle could induce freezing in the thin film, which could then trigger freezing in the seemingly detached secondary drop. Mechanical agitation or shock may also play a role in the freezing of the secondary drops (Alkezweeny, 1969, Czys, 1989). Regardless of the freezing mechanism, the glass slide will likely have some influence, and it will be pertinent to remove this in future investigations.

**RC:** Perhaps mechanical agitation could trigger a freezing event. There are reports in the literature of freezing catalyzed by collisions. (See Alkwezeeny 1969 and Czys 1989.)
**AC:** Added following sentence:
'Alternatively, mechanical agitation or shock may play a role in the freezing of the secondary drops (Alkezweeny, 1969; Czys, 1989).'

**RC:** To re-emphasize my earlier point… I am in favor of publication of this paper, despite my misgivings about some of the interpretation of the data. Secondary ice is an important topic, and I think we need to consider a very wide range of possibilities and mechanisms. Also, to be clear, I'm not asking for more experiments for this paper. Some clarification on the points I raised above would be good, but I think it is enough to acknowledge these points in the present work and leave further work for further work.
**AC:** We thank the referee for their comments and suggestions and have added clarification to the points raised above.

**Minor points:**

**RC:** The authors note that the falling drops were all the same size and fell from the same height, so that the impact velocity was 5.2 m/s. A comment here on how representative that might be as a closing velocity in the atmosphere (where it is most likely ice overtaking more slowly falling liquid drops) is warranted.
**AC:** Added:

'The terminal velocity of a 5 mm diameter drop is approximately 9 m s$^{-1}$ (Gunn 1949). Initially, the impact velocity may seem unrealistic. However, the ice particle in these experiments was held stationary on a glass slide, but in the atmosphere the ice particle would also be falling. The terminal velocity will depend on the ice particle shape, but for aggregates of similar size it is typically around 1 m s$^{-1}$ (Locatelli and Hobbs, 1974). The differential velocity between the supercooled water drop and ice particle will be less than 9 m s$^{-1}$ dependent on the nature of the ice particle. While such large droplets are rare in the atmosphere the purpose here is to demonstrate that the process is a potential secondary ice mechanism.'

**RC:** Line 176: 11 C. Missing a negative sign here?
**AC:** Yes – added negative sign.

**RC:** Final sentence of the paper: "Further work is needed..." I agree. This manuscript is an interesting addition to the literature in my opinion, but opens a lot of questions as well. (Many of the best papers do…)
**AC:** We agree that this work raises many more questions and thank the referee again for their constructive review! In light of another reviewer's comments, we have expanded on the conclusions section to include some suggestions on how we would further this work.

'One of the main experimental challenges of this work was dropping the supercooled water drop consistently onto the ice particle which limited the amount of experiments we could perform. As shown in Table A1, the majority of the successful impacts were classified as partial hits despite the intention for them to be direct hits. While partial hits are expected in clouds, as well as direct hits, we also conducted many experiments where the supercooled water drop missed the ice particle. One method of achieving better control of the supercooled water drop impact could be via growth and supercooling of a water drop at the end of a needle similar to the system shown in Schremb et al. (2018). Compared to our current mechanism, which involved tilting a pipette to allow the supercooled water drop to roll off, the supercooled water drop would remain fixed to a certain point before detaching under gravity, making it easier to drop consistently in the same position.

Another experimental challenge we would like to address is quantifying the secondary drops formed during the spreading phase of the supercooled water drop during impact. Thoroddsen et al. (2012) quantified secondary drops ejected with velocities of up to 100m s$^{-1}$ using an ultra-high-speed camera capable of recording at 1000000 fps, and we could use a similar setup. We could then exploit the birefringent properties of ice to determine whether these ejected secondary drops froze.

The number of secondary drops per collision is sensitive to geometry and material of collision, even for drops of the same size. We quantify about 10 per collision, Schremb et al. (2018) observed 10s of collision for impacts on elevated ice surface, Rozhkov et al. (2002) observe 100s for drop impacts on steel disks at room temperature, as do Villermaux and Bossa (2011) for drop impacts on iron cylinders at room temperatures. Consequently, after addressing the above challenges and elevating the ice particle off the glass surface, which may be achieved simply by fixing the ice particle on a wire, further work is needed to investigate, more systematically, this new SIP mechanism over a range of experimental parameters, not limited to: supercooled drop sizes, supercooled water drop-to-ice particle size ratios, ice particle shapes, temperatures, drop height (and hence impact velocity), airflow, relative humidity conditions and chemical compositions of the supercooled water drop.'

**References**

Alkezweeny, Freezing of supercooled water droplets due to collision. J. Appl. Meteorol. 1969, 8, 994–995.

Czys, Ice initiation by collision-freezing in warm-based cumuli. J. Appl. Meteorol. 1989, 28, 1098–1104.

Gunn and Kinzer, The terminal velocity of fall for water droplets in stagnant air, J. Atmos. Sci., 1949, 6, 243 – 24.

Locatelli, J. D. and Hobbs, P. V.: Fall speeds and masses of solid precipitation particles, J. Geophys. Res., 1976, 79,2185–2197.

Macklin and Payne, A theoretical study of the ice accretion process, Q. J. R. Meteorol. Soc., 1967, 93, 195–213.

Phillips et al., Secondary Ice Production by Fragmentation of Freezing Drops: Formulation and Theory, J. Atmos. Sci., 2018, 76, 3031–3070.

Pruppacher, H.R. and Klett, J.D., 2012. Microphysics of Clouds and Precipitation: Reprinted 1980. Springer Science & Business Media.

Rozhkov et al., Impact of water drops on small targets, Phys. Fluids, 2002, 14, 3485-3501

Schremb et al., Normal impact of supercooled water drops onto a smooth ice surface: experiments and modelling, J. Fluid Mech., 2018, 835, 1087–1107.

Thoroddsen et al., Micro-splashing by drop impacts, J.  Fluid Mech., 2012, 706, 560–570.

Villermaux and Bossa, Drop fragmentation on impact, J. Fluid Mech., 2011, 668, 412—-435.

---

## Author Comment (AC2)

**AC Response to RC2**

We thank the referee for their time in reviewing our manuscript and appreciate the constructive feedback given.

**RC:** I enjoyed reading this paper, which presents significant new experimental results relating to secondary ice processes, and is certainly worth publishing. I have a few minor questions and suggestions for making the paper a bit stronger (see below).

Active between -3 °C ≤ T ≤ -8 °C, rime–splintering occurs when supercooled water drop diameters are < 13 μm or > 24μm (Hallett and Mossop, 1974; Mossop and Hallett, 1974; Mossop, 1978) Maybe I misremember the Mossop 1978 paper, but should the condition on droplet sizes here be "< 13 μm and> 24μm" rather than or?

**AC:** Changed.

Section 2

**RC:** Perhaps it could be useful just to elaborate a little bit on these formulae. Maybe showing a figure with graphs of Ns vs DE; Phi vs T; f(T) would help make the general behaviour clearer?

**AC:** We have removed Section 2 (Theory) due to comments from another reviewer, so the theory is no longer described in this paper.

**RC:** The typical freezing shape of the ice particle is shown in Fig. 1.

Is the shape of the ice particle likely to be a relevant factor here? How could you find out? What is it likely to be in the atmosphere?

**AC:** Added section in discussion:

'Another factor that will influence the generation of secondary drops is the ice particle shape. Currently, our ice particles have a pointed tip, as shown in Fig. 1, which is a typical shape formed when a liquid water drop is frozen on a cold substrate (Snoijer et al.,2012), but not representative of atmospheric ice particles. According to Phillips et al. (2018), who refer to this SIP mechanism `Mode 2', for it to occur, the supercooled water drops must have a diameter larger than 150 μm and the ice particle more massive still. In the atmosphere, ice particles which are larger than 150 μm are typically irregular in shape (Korolev and Sussman, 2000). A study by Zhang et al. (2020) shows that at room temperature, water drop impact on curved surfaces induce additional fragmentation mechanisms compared to flat surfaces. Therefore, we expect the irregular shape of an ice particle to affect the fragmentation mechanisms of the supercooled water drop and thus secondary drop formation. Exactly how irregular particle shapes will change the secondary drop formation is difficult to ascertain without further studies.'

**RC:** the impact velocity (V0) for all experiments was 5.2m s−1

it's worth pointing out in the text that this is below terminal velocity for a 5mm drop(which would be closer to 9 m/s). However in the real atmosphere the ice particle would be moving as well, so the differential velocity may be more realistic than it might initially appear.

**AC:** We've added the following sentences in Section 2:

'The terminal velocity of a 5 mm diameter drop is approximately 9 m s$^{-1}$ (Gunn 1949). Initially, the impact velocity may seem unrealistic. However, the ice particle in these experiments was held stationary on a glass slide, but in the atmosphere the ice particle would also be falling. The terminal velocity will depend on the ice particle shape, but for aggregates of similar size it is typically around

1 m s$^{-1}$ (Locatelli and Hobbs, 1974). The differential velocity between the supercooled water drop and ice particle will be less than 9 m s$^{-1}$ dependent on the nature of the ice particle.

RC: In fluid dynamics, the Weber number, We = $\rho DV0^2/\sigma$, and Reynolds number, Re = $\rho DV0/\mu$, are used to relate inertial forces to interfacial and viscous forces, respectively. Taking into account the temperature dependent values of surface tension and viscosity of the supercooled water between -4 °C ≤ T ≤ -12 °C, the We and Re number ranges obtained were 1747 ≤ We ≤ 1772 and 8781 ≤ Re ≤ 12240, respectively.

I think in both cases here, it would be good to clarify what We and Re refer to–or more specifically where these inertial, viscous, and interfacial forces are acting. Often in cloud physics we think about the inertial, viscous in the air surrounding the drop, while here (I think) you are considering them within the water

AC: We've added the following words highlighted in bold:

'…used to relate inertial forces of the **fluid** to its interfacial and viscous forces respectively.

And, added the following sentences to clarify:

'In this case, the fluid is the supercooled water drop. The inertial force is from the initial impact velocity of the supercooled water drop, and the interfacial (surface tension) and viscous forces are properties of the supercooled water drop.'

RC: Is it obvious what the length scale and velocity scale in We and Re should be? You have chosen V0 for the velocity scale, so that implies the water fluid parcels of interest are moving at this velocity. So are you considering the downward motion of the liquid water at the moment of impact on the ice particle? Or the lateral velocity of the liquid water as it spreads out? (are these velocity scales comparable?).

AC: In drop impact experiments it is typical that the length scale should the diameter of the initial drop before impact and the velocity scale is the normal impact velocity of the drop on the surface (e.g., see a review of drop impact by Josserand & Thorodssen 2016). So yes, this is the downward motion of the liquid water at the moment of impact on the ice particle, not the lateral velocity as the liquid water as it spreads out.

We have added the following to word in bold to clarify:

'…**initial** supercooled water drop diameter **before impact** (D)'

'…the **normal** impact velocity (V0)…'

It's difficult to say whether the impact and lateral velocity scales are comparable. Zhang et al. (2021) show in their Fig 9(a) that there is a linear relationship between the impact velocity and the lateral (spreading) velocity for water drops on flat surfaces with varying degrees of wettability. For a superhydrophobic surface (similar to our glass slide) with an impact velocity of 5 m/s the spreading velocity is ~9 m/s. However, our supercooled water spreads over an ice particle and the glass slide which will likely reduce the velocity.

RC: For the length scale, it's not obvious what to choose, when you have a liquid spreading over a solid surface. The depth of the water coating? D is probably not an unreasonable choice, but maybe you can make the argument a bit more explicit somehow. Again, it all comes down to what aspect of the flow of the water you are trying to characterise.

AC: The length scale typically used in drop dynamics is the diameter of the initial drop before impact. See response to question above for clarification made in the text.

RC: Section 3 –you used a high speed camera. What exposure time was used? It seems from the images like the splash itself (t=0) is quite blurred. Was this limited by the illumination?

**AC:** The exposure time was 929.36 μs. We believe the frames appear blurred because of difficulties in knowing where the drop would impact, which made it difficult to focus the camera.

**RC:** For figs 2,3,4 I did wonder whether adding some slightly more detailed description of what's happening in the various frames would help the reader interpret what they are seeing. It took me a while to get a sense of what was happening. Or maybe some extra annotation on the figures themselves?

**AC:** We've annotated Fig 3 (originally Fig 2) and Fig 4 (originally Fig 3) with letters to indicate the impact phase, spreading phase, secondary drop formation/ejection during the spreading phase, retraction phase, secondary drop formation due to receding break-up and partial rebound. Plus a clarifying sentence highlighted in yellow in the caption. See below:

[Figure]

**Figure 3.** Frames from the high–speed camera configuration of a water drop impact on a glass slide when both water drop and glass slide are at (a) room temperature (23 °C) and (b) -5 °C. The impact phase (I), spreading phase (S), secondary drop formation/ejection during the spreading phase (E), retraction phase (R), secondary drop formation due to receding break-up (B) and partial rebound (PR) of the water drop are indicated in the frames. Arrows indicate secondary drop formation during the retraction phase of the water drop.

[Figure]

**Figure 4.** Frames from the high–speed camera configuration of a supercooled water drop impact on an ice particle when both drop and ice particle are at -5 °C. The impact phase (I), spreading phase (S), secondary drop formation/ejection during the spreading phase (E), retraction phase (R), secondary drop formation due to receding break-up (B) and partial rebound (PR) of the water drop are indicated in the frames. Arrows indicate secondary drop formation during the retraction phase of the supercooled water drop.

In Fig 4 we've added before impact, near impact, ~10s after impact and difference between before and after impact to the frames. We've also indicated that the top panel is from the RPicam with no polarising filter showing both liquid and solid phase water and that the bottom panel is from the RPicam with a polarising filter showing ice only. See below:

[Figure]

**Figure 5.** Selected frames from the impact of a supercooled water drop on an ice particle at -4°C using the RPicams configuration. Frames (a)–(c) before, at and ~10 s after impact using the camera with no polarising filter. Red arrows in (c) indicate the number of secondary drops formed. Frame (d) shows the difference between (a) and (c). Frames (e)–(g) before, at and ~10 s after impact using the camera with a polarising filter. The white arrow in (h) indicates the frozen secondary drop. Frame (h) shows the difference between (e) and (g).

**RC:** Discussion -You mention the influence of the glass slide, and I agree the presence of the slide itself is definitely worth discussing. Another factor I can think of here is that the ice particle is effectively in a fixed vertical position, while in the atmosphere the ice particle is in free fall, and when the drop hits it, then the ice particle can move in response to that –so some of the drop's momentum can be carried to the ice particle. Would that change the way the water flows over the ice particle, and freezes?

**AC:** We thank the reviewer for making this good point. We think that the ability of the ice particle to move upon collision will have some effect on the way it fragments, and it is certainly something we would like to explore in the future. However, we don't really know whether it will increase or decrease the fragmentation of the supercooled water drop without first investigating the fragmentation mechanism of the supercooled water drop without the glass slide. We know from a study by Zhang et al. (2020) that curved surfaces can cause additional fragmentation mechanisms. Another factor will be the freezing mechanism which we also need to investigate further. If freezing is initiated by the formation of the ice dendrites from contact with the ice particle, which can occur on the millisecond scale, then the ice dendrites will still be able to propagate through the supercooled water drop even if contact time is reduced. If freezing is via mechanical agitation/shock then the momentum transfer to the ice particle from the supercooled water drop will likely have a more significant effect on freezing. We have added the following paragraph below:

'In addition, the ice particle in our experiments is in a fixed position on the glass slide, whereas, in the atmosphere, the ice particle is in free fall. When the faster-moving supercooled water drop collides with the ice particle, the ice particle will move in response to the collision, likely affecting the formation of the secondary drops and their subsequent freezing. However, currently, it is difficult to ascertain how this will influence secondary drop formation and freezing without further investigations into the mechanisms of secondary drop formation on an elevated ice particle.'

**RC:** In figure 5 I think it's important to clarify what the error bars represent in the caption, and in the text. Is it the variation from one experiment to the next, in the "same" conditions? Or is it the uncertainty on the mean value?
**AC:** Added the following to caption:
'The error bars represent the standard error in the temperature intervals which are listed in Table A2 & A3.'

**RC:** Connected to this is Table A2 –the values of phi, sigma, and sigma_phi_bar are all quoted to the nearest 0.1, which seems a bit coarse. Might be worth 1 extra significant figure?
**AC:** Added another significant figure and updated Fig. 5 to reflect this change. See below:

Table A2:

| T interval (°C) | | $\bar{T}$ (°C) | $\bar{\Phi}$ | $\sigma$ | n | $\sigma_{\bar{\Phi}}$ |
|---|---|---|---|---|---|---|
| -3.8 | -4.3 | -4.1 | 0.38 | 0.26 | 5 | 0.12 |
| -5.3 | -5.8 | -5.6 | 0.36 | 0.34 | 5 | 0.15 |
| -6.0 | -6.1 | -6.1 | 0.44 | 0.38 | 4 | 0.19 |
| -7.7 | -7.8 | -7.8 | 0.29 | 0.17 | 2 | 0.12 |
| -8.0 | -8.5 | -8.2 | 0.19 | 0.30 | 4 | 0.15 |
| -9.4 | -9.9 | -9.7 | 0.22 | 0.19 | 3 | 0.11 |
| -10.0 | -10.1 | -10.1 | 0.80 | 0.28 | 2 | 0.20 |
| -11.3 | -11.9 | -11.7 | 0.25 | 0.00 | 2 | 0.00 |

Figure 5:

[Figure]

**RC:** The number of experiments is fairly small, given the variability in phi that's shown. I'm guessing these are quite time consuming to conduct and analyse. Perhaps you can discuss that a bit? In general I would enjoy seeing an expansion of the future work in section 6 to talk about how the experiment could be improved and elaborated. Likewise saying "no quantification of the freezing fraction of the secondary ice drops [from the jet of smaller droplets] can currently be made" is fine, but it would be good to discuss what you would need to do to quantify it, or study it in more detail.
**AC:** We have added the following paragraphs:

'One of the main experimental challenges of this work was dropping the supercooled water drop consistently onto the ice particle which limited the amount of experiments we could perform. As shown in Table A1, the majority of the successful impacts were classified as partial hits despite the intention for them to be direct hits. While partial hits are expected in clouds, as well as direct hits, we also conducted many experiments where the supercooled water drop missed the ice particle. One method of achieving better control of the supercooled water drop impact could be via growth and supercooling of a water drop at the end of a needle similar to the system shown in Schremb et al. (2018). Compared to our current mechanism, which involved tilting a pipette to allow the supercooled water drop to roll off, the supercooled water drop would remain fixed to a certain point before detaching under gravity, making it easier to drop consistently in the same position.

Another experimental challenge we would like to address is quantifying the secondary drops formed during the spreading phase of the supercooled water drop during impact. Thoroddsen et al. (2012)

quantified secondary drops ejected with velocities of up to 100m s$^{-1}$ using an ultra-high-speed camera capable of recording at 1000000 fps, and we could use a similar setup. We could then exploit the birefringent properties of ice to determine whether these ejected secondary drops froze.

The number of secondary drops per collision is sensitive to geometry and material of collision, even for drops of the same size. We quantify about 10 per collision, Schremb et al. (2018) observed 10s of collision for impacts on elevated ice surface, Rozhkov et al. (2002) observe 100s for drop impacts on steel disks at room temperature, as do Villermaux and Bossa (2011) for drop impacts on iron cylinders at room temperatures. Consequently, after addressing the above challenges and elevating the ice particle off the glass surface, which may be achieved simply by fixing the ice particle on a wire, further work is needed to investigate, more systematically, this new SIP mechanism over a range of experimental parameters, not limited to: supercooled drop sizes, supercooled water drop-to-ice particle size ratios, ice particle shapes, temperatures, drop height (and hence impact velocity), airflow, relative humidity conditions and chemical compositions of the supercooled water drop.'

**References**

Gunn and Kinzer, The terminal velocity of fall for water droplets in stagnant air, J. Atmos. Sci., 1949, 6, 243 – 24.

Josserand and Thoroddsen, Drop Impact on a Solid Surface, Annu. Rev. Fluid Mech., 2016, 48, 365-391.

Korolev and Sussman, A Technique for Habit Classification of Cloud Particles, J. Atmos. Ocean. Technol., 2000, 17, 1048 – 1057.

Locatelli, J. D. and Hobbs, P. V.: Fall speeds and masses of solid precipitation particles, J. Geophys. Res., 1976, 79,2185–2197.

Phillips et al., Secondary Ice Production by Fragmentation of Freezing Drops: Formulation and Theory, J. Atmos. Sci., 2018, 76, 3031–3070.

Rozhkov et al., Impact of water drops on small targets, Phys. Fluids, 2002, 14, 3485-3501

Schremb et al., Normal impact of supercooled water drops onto a smooth ice surface: experiments and modelling, J. Fluid Mech., 2018, 835, 1087–1107.

Snoijer and Brunet, Pointy ice-drops: How water freezes into a singular shape, Am. J. Phys., 2012, 80, 764–771.

Thoroddsen et al., Micro-splashing by drop impacts, J. Fluid Mech., 2012, 706, 560–570.

Villermaux and Bossa, Drop fragmentation on impact, J. Fluid Mech., 2011, 668, 412—-435.

Zhang et al., Asymmetric splash and breakup of drops impacting on cylindrical superhydrophobic surfaces, Phys. Fluids, 2020, 32, 122 108.

Zhang et al., Effect of wettability on droplet impact: Spreading and splashing, Exp. Therm. Fluid Sci., 2021, 124, 110369.

---

## Author Comment (AC3)

**AC Response to RC3**

We thank Sylvia for taking the time to review our manuscript and appreciate the constructive feedback given.

**RC:** James, Phillips, and Connolly present experimental results on a secondary ice production (SIP) process involving liquid drop-ice crystal collisions. It is nice to see additional, and especially quantitative, laboratory results on SIP, and I support publication of the article. I feel, however, that several points should be elaborated and that some reorganization of sections would help with clarity.

**Major Comments**

- Given that a condensed version of the theoretical work in Phillips et al. 2018 was presented, I expected there to be some comparison of the observational results with this model. For example, can you calculate Ns as a function of temperature from the Phillips et al. model and overlay it on Figure 5? Can you say anything about the validity / assumptions of the model on the basis of these experiments?

**AC:** The figure below shows a comparison of the model with our experimental results.

[Figure]

We have decided to remove Section 2 as we cannot currently quantify the majority of the secondary drops formed in our experiments. Therefore, we believe a comparison between our experimental results and the model presented in Phillips et al. (2018) does not provide anything beneficial at this time. We would like to provide quantification of the majority of secondary drops in the future through additional experiments and these will be compared with the model understand the validity/assumptions of the model.

**RC:** - I felt that the mechanistic discussion at the start of Section 5 would have been helpful prior to Sections 4.1 and 4.2, so that I had a better sense of what was physically happening in the experiments in Figs 2 and 3.

**AC:** Changed. We have also added some more detail about the drop impact phase in Figs 3 (originally 2) and 4 (originally 3). See below.

[Figure]

**Figure 3.** Frames from the high–speed camera configuration of a water drop impact on a glass slide when both water drop and glass slide are at (a) room temperature (23 °C) and (b) -5 °C. The impact phase (I), spreading phase (S), secondary drop formation/ejection during the spreading phase (E), retraction phase (R), secondary drop formation due to receding break-up (B) and partial rebound (PR) of the water drop are indicated in the frames. Arrows indicate secondary drop formation during the retraction phase of the water drop.

[Figure]

**Figure 4.** Frames from the high–speed camera configuration of a supercooled water drop impact on an ice particle when both drop and ice particle are at -5 °C. The impact phase (I), spreading phase (S), secondary drop formation/ejection during the spreading phase (E), retraction phase (R), secondary drop formation due to receding break-up (B) and partial rebound (PR) of the water drop are indicated in the frames. Arrows indicate secondary drop formation during the retraction phase of the supercooled water drop.

**RC:**- I appreciate that the limitation of the glass slide is acknowledged and secondary drop production in other setups is discussed (lines 159-171). But does the presence of the glass slide also mean that the mechanism shown in Figure 6 may need to be modified in the real atmosphere?

**AC:** Figure 6 (now Figure 2) represents a schematic of the experiments and is not representative of the process which would likely occur in the atmosphere. We have added the following words in bold in the caption to clarify:

'A schematic diagram of a supercooled water drop impact on an ice particle **on a glass slide**…'

**RC:** In particular, given that the secondary drops and frozen fraction have only been quantified in the retraction phase, should an equivalent retraction phase exist between two curved surfaces in relative motion?

**AC:** A retraction phase occurs over stationary curved surfaces, e.g. see Zhang et al. (2020). In fact, a curved surface is more likely to induce additional fragmentation mechanisms compared to a flat surface. In the atmosphere the ice particles are likely to be irregular in shape. We have included the below paragraph discussing the role of the shape of the ice particles:

'Another factor that will influence the generation of secondary drops is the ice particle shape. Currently, our ice particles have a pointed tip, as shown in Fig. 1, which is a typical shape formed when a liquid water drop is frozen on a cold substrate (Snoijer et al.,2012), but not representative of atmospheric ice particles. According to Phillips et al. (2018), who refer to this SIP mechanism `Mode 2', for it to occur, the supercooled water drops must have a diameter larger than 150 μm and the ice particle more massive still. In the atmosphere, ice particles which are larger than 150 μm are typically irregular in shape (Korolev and Sussman, 2000). A study by Zhang et al. (2020) shows that at room temperature, water drop impact on curved surfaces induce additional fragmentation mechanisms compared to flat surfaces. Therefore, we expect the irregular shape of an ice particle to affect the fragmentation mechanisms of the supercooled water drop and thus secondary drop formation. Exactly how irregular particle shapes will change the secondary drop formation is difficult to ascertain without further studies.'

In relative motion we would still expect to see secondary drop formation during the retraction phase as surface tension is responsible for the retraction of the drop. We do want to pursue investigating more realistic supercooled water drop collisions and ice particles in the future.

RC:- Section 5.1 read like introductory material, as it did not include discussion of any of the study's findings. It is valuable information, but I would integrate it into the introduction as general motivation to study SIP.
AC: We have integrated it into the introduction.

RC:- In place of Section 5.1, I think a new (small) "Atmospheric Implications" section or additional sentences throughout the Discussion / Conclusion would be helpful to discuss the atmospheric conditions under which the experiments are representative, i.e. In what regions / synoptic conditions / cloud systems, would ice particles of diameter 6 mm and raindrop of diameter 5 mm coexist? For what range of ice particle and raindrop sizes, does 5.2 m s-1 represent a realistic relative terminal velocity? Would it be possible to use the We and Re characterizing these experiments to identify regimes in in-situ data for which this mode 2 fragmentation could occur? etc.
AC: We don't think that supercooled water drops with diameters of 5 mm and ice particles with diameters of 6 mm are necessarily representative of cloud conditions. Practically, it was easier for us to work with supercooled water drops of this size. The intention of this paper is to demonstrate that the collisions of supercooled water drops with ice particles could be a new SIP. We have removed the Atmospheric Implications section and added the following paragraph at the end of the Discussion session.

'As a proof-of-concept investigation, we studied supercooled water drops with diameters of 5 mm and ice particles with diameters of 6 mm as larger sizes of supercooled water drops were easier to work with experimentally. While these sizes are not necessarily representative of cloud conditions, theoretically, this new SIP mechanism should occur where supercooled water drop diameters are > 150 μm and the ice particles more massive still. Supercooled water drops and ice particles are present within a variety of different clouds. For example, Hobbs and Rangno (1990) presented aircraft observations in small polar--maritime cumuli that displayed ice enhancement. Their discussion highlighted that ice enhancement proceeded in two stages. The first stage consisted of the formation of frozen drops, < 400 μm diameter, and small graupel particles, < 1 mm diameter. The second stage was characterised by the appearance of high concentrations of vapour--grown ice crystals in the upper regions of the cloud. A key finding of this series of papers was that high concentrations of small ice particles appeared simultaneously with frozen drizzle drops. Furthermore, Rangno and Hobbs (2001) showed that large supercooled drops were often a requirement for ice enhancement in moderately cooled Arctic stratiform clouds, and ice enhancement was often coincident with observations of large supercooled raindrops. Supercooled drizzle drops and raindrops are common in convective clouds (e.g. Crawford et al., 2012, Taylor et al., 2016), as are large ice particles. Hence, because there is a broad continuum of drizzle and raindrop sizes, where the larger drops freeze first, followed by accretion of the smaller unfrozen drops that the collision of supercooled water drops with ice particles more massive may be of importance in a wide range of clouds.'

**Minor Comments**

RC: Line 15 – "where subzero temperatures"
AC: Added.

**RC:** Lines 17-18 – "typically fall between 1 x 10-5 L-1 and 1 L-1 at temperatures T~-10 deg C" (looking at Fig. 1-10 from Kanji et al. 2017)
**AC:** Changed.

**RC:** Line 24 – "NWP models underestimate the concentrations of ice particles" – It would be nice to include an order-of-magnitude range for these underestimates.
**AC:** Upon reflection we think that this is probably a too broad a statement about NWP models underestimating ice particle concentrations and have removed it. Some NWP model can get the right order of magnitude, but sometimes for the wrong reasons. They often use parameterisations that generate too many crystals by primary ice nucleation (e.g. the Cooper, 1986 nucleation description), when secondary ice may be responsible. The issue with over estimating primary IN is that it then leads to a glaciation of the clouds and underestimation of supercooled liquid water, which then leads to an underestimation of secondary ice mechanisms that rely on supercooled water being present (there are other reasons for the underestimation of supercooled liquid water too, related to the resolution of the models). There are instances where NWP models underestimate ice particle concentrations, such as in Crawford et al. (2012), where observed concentrations of were ~ 100/L ice particles, but the NWP model, Weather, Research and Forecasting, estimates only up to 30 /L, and more broadly 5/L.

**RC:** Line 28 – "supercooled water drop diameters are < 13 um and > 24 um" In Hallett and Mossop 1974, both droplet sizes should coexist.
**AC:** Changed.

**RC:** Line 31 – Along with the temperature range for frozen droplet shattering, it would be worthwhile to include a droplet size range as well, since droplet size will be discussed later as a parameter of the current experiments (e.g. 280-350 um in Keinert et al. 2020)
**AC:** Added the following:
'A range in diameters of freezing supercooled water drops has also been investigated between laboratory studies from 4 μm to 1000 μm (see Table 1 of Korolev and Leisner, 2020, for a summary).'

**RC:** Line 34 – I would suggest to rephrase as "the attention of laboratory studies has overwhelmingly focused on the rime-splintering...", since a growing body of recent work has look at breakup parameterizations (e.g. Hoarau et al. 2018, Sotiropoulou et al. 2020, Sotiropoulou et al. 2021, Dedekind et al. 2021, etc.)
**AC:** Changed.

**RC:** Line 52 – If you note that 'Mode 2' of frozen droplet fragmentation is studied, it would be helpful to know what 'Mode 1' is also.
**AC:** We have removed Section 2 so this is no longer in the paper. However, we have clarified 'Mode 1' and 'Mode 2' where the first reference to Phillips et al. (2018) is made:
'This SIP mechanism has been investigated via a theoretical study by Phillips et al. (2018) and referred to as 'Mode 2' as it involves collisions of supercooled water drops with more massive ice particles resulting in fragmentation of the supercooled water drop. Ice contained in some of the secondary drops was assumed to initiate freezing, yielding secondary ice fragments. By contrast, 'Mode 1' involved either collisions of supercooled water drops with less massive ice particles resulting in spherical freezing of the supercooled water drop or activation of immersed INPs, with a quasi-spherical outer ice shell that fragments. '

**RC:** Line 64 - "DEcrit" (not Dcrit). Also you have not yet defined the freezing stages when you mention "stage1 of freezing" here and again in Line 68.
**AC:** Thank you. We have removed Section 2 so this is no longer in the paper.

**RC:** Line 70 – "Finally, Phillips et al. 2018 hypothesised that $\Phi(T) = min[4f(t),1]$ such that $\Phi = 0.5$ at -10 C" Stated like this, it sounds rather ad hoc. Perhaps an additional sentence can clarify where this form comes from or how it is constrained.
**AC:** Thank you. We have removed Section 2 so this is no longer in the paper.

**RC:** Line 94 – I do not know how important it is, but it was not clear to me what the "x-y translator (modified 3D printer)" was in the setup.
**AC:** The pipette could be moved to different positions using an x-y translator so that multiple drop collision experiments could be performed on one glass slide as, at the time, we didn't have access to multiple glass slides of the correct size. In hindsight, we would opt for a fixed design. This detail is not important to the experiment, so we have removed mention of it from the text and simplified Fig. 1 to avoid any confusion. See below:

[Figure]

**RC:** Line 99 – Surface tension of the liquid water is presented as γ in line 61 and σ here; viscosity is presented as λ here and μ in the equation for the Reynolds number.
**AC:** Changed surface tension to σ throughout text and viscosity to μ.

**RC:** Lines 113-114 – It was not clear to me why the filament-like structures do not form when the colliding droplet spreads on the glass slide at room temperature. Is there a physical explanation for why this only occurs at colder temperatures?
We are also curious about why the subzero glass slide experiments exhibit filament-like structures upon impact. Drop impact on superhydrophobic surfaces at room temperature do not usually

appear to form filaments. As the temperature of water decreases the viscosity and surface temperature change so the appearance of filaments at supercooling could be something to do with the change in properties of supercooled water.

**RC:** Section 4.1 – Somewhere in this section or perhaps in the preceding Section 3, it would be helpful to have already referred to Table A1, so that it is clear from how many experiments the results come, e.g. only two glass slide collisions total were performed at 23 and -5 C?
**AC:** We only conducted a few experiments on the bare glass slide using the high-speed camera at 23 and -5 °C as this was for qualitative purposes to gain an understanding of the water drop fragmented and how the ice particle influenced this fragmentation. We have added the following sentence at the end of Section 3:
"We conducted 32 experiments using the RPicams configuration during quantification of the freezing fraction of secondary drops and the data is given in Table A1."

**RC:** Lines 133-134 – Was anything learned from the partial versus direct collisions? Does one or the other produce more secondary drops or higher frozen fraction? I guess there may be no robust difference, given the difficulty of performing these direct collisions.
**AC:** With the current dataset, we didn't observe a discernible difference between partial and direct collisions. It is something we would like to investigate further by adapting the setup to make it easier to get more direct collisions. We have expanded on this at the end of the conclusions with the following paragraph:

'One of the main experimental challenges of this work was dropping the supercooled water drop consistently onto the ice particle which limited the amount of experiments we could perform. As shown in Table A1, the majority of the successful impacts were classified as partial hits despite the intention for them to be direct hits. While partial hits are expected in clouds, as well as direct hits, we also conducted many experiments where the supercooled water drop missed the ice particle. One method of achieving better control of the supercooled water drop impact could be via growth and supercooling of a water drop at the end of a needle similar to the system shown in Schremb et al. (2018). Compared to our current mechanism, which involved tilting a pipette to allow the supercooled water drop to roll off, the supercooled water drop would remain fixed to a certain point before detaching under gravity, making it easier to drop consistently in the same position.'

**RC:** Also was Figure 5 produced from all data (both partial and direct) in Table A1? This should be specified in the caption.
**AC:** Figure 5 was produced from all data. We have added the following to Fig. 5 caption: "Average data included both direct and partial collisions."

**RC:** Lines 136-138 – "The smaller secondary drops observed at impact ... were not observed." This seems like it may be an important limitation. Is there the possibility to improve RPicam resolution in future work? This should be mentioned in the conclusions / future work if so.
**AC:** We would more likely try to quantify the smaller secondary drops formed during impact/spreading phase with the high-speed camera. We have added the following sentences in the conclusions/future work:

'Another experimental challenge we would like to address is quantifying the secondary drops formed during the spreading phase of the supercooled water drop during impact. Thoroddsen et al. (2012) quantified secondary drops ejected with velocities of up to 100 m s$^{-1}$ using an ultra-high-speed camera capable of recording at 1000000 fps, and we could use a similar setup. We could then exploit the birefringent properties of ice to determine whether these ejected secondary drops froze.'

**RC:** Line 148 – "Surface tension and viscosity forces were considered negligible during the spreading phase of the drop" I am confused by this statement. Where / in which calculations are these forces being considered negligible?

**AC:** These forces are considered in the Weber and Reynolds number calculations given in the second to last paragraph of Section 2. The Weber number relates the inertia to surface tension, and the Reynolds number relates inertia to viscosity. When We and Re numbers are over a critical value then inertia is the dominating force during spreading, and surface tension and viscosity are considered negligible.

We've changed the sentence to: "Surface tension and viscosity forces were therefore considered negligible during the spreading phase of the drop"

**RC:** Lines 150-151 – I have not seen the prompt-type / corona-type splash terminology before; I would define these terms more completely from the citations in these lines.

**AC:** In prompt splashing, secondary drops are formed from the break-up of the advancing thin film. Whereas, in corona splashing the thin film forms a bowl-like structure which then breaks up to form secondary drops. In general it is difficult to discern between these mechanisms (see paper by Josserand and Thoroddsen, 2016) and our set-up is not designed to study this splashing mechanism. As adding this terminology will probably create more confusion, we have decided to remove this sentence from the manuscript.

**RC:** Figure 7 – Are the top versus bottom panels also with and without the polarising filter?

**AC:** Yes. We've added the following sentence in the caption:

"The top panel shows frames from the RPicam with no polarising filter and the bottom panel shows frames from the RPicam with a polarising filter."

[Figure]

**Figure 5.** Selected frames from the impact of a supercooled water drop on an ice particle at -4°C using the RPicams configuration. Frames (a)–(c) before, at and ~10 s after impact using the camera with no polarising filter. Red arrows in (c) indicate the number of secondary drops formed. Frame (d) shows the difference between (a) and (c). Frames (e)–(g) before, at and ~10 s after impact using the camera with a polarising filter. The white arrow in (h) indicates the frozen secondary drop. Frame (h) shows the difference between (e) and (g).

**RC:** Line 176 – "T less than equal to -11 deg C"

**AC:** Added.

**RC:** Lines 180-187 – I find the arguments here difficult to follow. The takeaway is that temperature dependence of frozen fraction is caused by a liquid-ice interaction time scale? Could the authors reword somehow for clarity?

**AC:** We've reworded the paragraph to the following:

'We observed a decrease in the number of secondary drops formed during receding break-up as

temperatures decreased below -8 ° C. Figure 7 shows the frames after a supercooled water drop impact with an ice particle for the experiments between -11 °C and -12 °C which was the range where the smallest number of secondary drops formed. At these temperatures, the supercooled water drop froze either during the spreading phase or in the early stages of the retraction phase. As the growth velocity of ice in supercooled water increases with decreasing temperature, e.g. at -2 °C it is around 0.2 cm s$^{-1}$, whereas at -10 °C it is around 5 cm s$^{-1}$ (see Pruppacher and Klett, 1997, chapter 16), which may explain why a decrease in secondary drops was observed. We believe the decrease in secondary drop formation at temperatures below -8 °C may be due to the artificially flat geometry presented by the glass slide and to the large size of the incident drop, both factors which prolonged the interaction time between the supercooled water drop and ice. For example, the supplementary videos from Schremb et al. (2018) showed several secondary drops forming at -14 °C after impact on an elevated ice target, more than we observed at our lowest temperature of -12 °C.'

RC: Line 189 – "We believe that the freezing fraction of the secondary drops is independent of the number of drops formed." Is there a reason for this belief? I would expect temperature dependence to dominate also, but I could also imagine that when a fixed fraction of the colliding droplet mass produces secondary drops, and more such secondary drops form, they are smaller and freeze faster..?
AC: We have removed this paragraph in part due to another referee's comment.

**References**

Cooper W.A. (1986) Ice Initiation in Natural Clouds. In: Precipitation Enhancement—A Scientific Challenge. Meteorological Monographs. American Meteorological Society, Boston, MA.

Crawford et al., Ice formation and development in aged, wintertime cumulus over the UK: observations and modelling, Atmos. Chem. and Phys., 2012, 12, 4963–4985,

Hobbs and Rangno, Rapid development of high ice particle concentrations in small polar maritime cumuliform clouds, J. Atmos. Sci., 1990, 47, 2710–2722.

Josserand and Thoroddsen, Drop Impact on a Solid Surface, Annu. Rev. Fluid Mech., 2016, 48, 365-391.

Korolev and Leisner, Review of experimental studies of secondary ice production, Atmos. Chem. Phys., 2020, 20, 11767–11797.

Phillips et al., Secondary Ice Production by Fragmentation of Freezing Drops: Formulation and Theory, J. Atmos. Sci., 2018, 76, 3031–3070.

Pruppacher, H.R. and Klett, J.D., 2012. Microphysics of Clouds and Precipitation: Reprinted 1980. Springer Science & Business Media.

Rangno, A. L. and Hobbs, P. V.: Ice particles in stratiform clouds in the Arctic and possible mechanisms for the production of high ice concentrations, J. Geophys. Res., 2001, 106, 15065–15075.

Schremb et al., Normal impact of supercooled water drops onto a smooth ice surface: experiments and modelling, J. Fluid Mech., 2018, 835, 1087–1107.

Taylor et al., Observations of cloud microphysics and ice formation during COPE, Atmos. Chem. Phys., 2016, 16, 799–826.

Thoroddsen et al., Micro-splashing by drop impacts, J. Fluid Mech., 2012, 706, 560–570.

---

## Referee Report (RR1)

***Review 2 of Secondary ice production during the break-up of freezing water drops on impact with ice particles***

Sylvia Sullivan

I appreciate the authors' efforts to revise the manuscript and find that I understand the proposed mechanism and setup much better now. For example, the paragraph in lines 35-41 is now quite helpful to understand the difference between mode 1 and mode 2 drop freezing and fragmentation, and addition of mechanism 'phase labels' to Figures 3 and 4 is nice. I also appreciate mention of experimental challenges and future work in Section 5. I have a series of minor comments and edits after my second read-through, but otherwise support publication of the manuscript.

**Comments**

Abstract - The abstract starts brusquely. I would add an initial contextual sentence about the importance of secondary ice processes and the contribution of these experiments to better understanding their mechanisms.

Lines 51-64 – These paragraphs seem to me to fit better after line 23. Then you have discussed the general importance of secondary ice, first to understand discrepancies between INP and ICNC and second to explain persistent generation of ice in thin mixed-phase clouds. After that, you present rime-splintering as the most widely employed and studied mechanism and finish by suggesting that this new mode 2 drop freezing and fragmentation could also be important.

Lines 95-96 – Is a more convincing argument here for the relevance of this impact velocity that the mixed-phase region, at least of deep convective clouds where liquid and ice hydrometeors grow to the large sizes here, are highly turbulent? And impact velocities within the turbulent eddies could be quite large?

Discussion – I would also add one introductory sentence here to help the reader orient, something like "We discuss some aspects of the experimental setup that may affect the occurrence and rate of secondary drop production and freezing here."

**Minor Edits**

Line 29 – I would remove "than rime-splintering", as the sentence reads more cleanly then. You could also specific that the variation in "quantification between laboratory studies" is quantification *of ice fragment generation rates and temperature dependence in these rates.*

Line 68 – drops *that* freeze

Line 98 – (Locatelli and Hobbs 1974) adjust parentheses

Line 106-107 – You already defined *D* within We and Re above, but it is perhaps worth reiterating here that you define the length scale in these dimensionless number to be the water drop diameter prior to impact.

Lines 131-133 – I asked why these filaments are only produced at colder temperatures, and I feel it would be worthwhile to mention the point you made about increased viscosity and surface tension of supercooled water explicitly, e.g. "where no ejection of filament-like structures was observed, *perhaps due to lower viscosity and surface tension of water at these temperatures*"

Figure 6 caption – "The error bars represent the standard error *in freezing fraction or secondary drop number for the* temperature intervals…" Quite minor but just for clarity

Line 167 – "on an elevated ice surface" From this description, it sounds like the setup in Schremb et al. 2018 also used a flat surface for impact (an icy one not glass). But later it is stated that "when a flat surface… is not present, secondary drops are still formed." Could you clarify?

Line 189 – "we expect the irregular shape of an ice particle to *enhance* the fragmentation mechanisms" ?

Line 246 – "freezing fraction of the secondary drops" Omit "*ice*", right?

Lines 261-262 – suggested rewording to avoid a run-on "We measure about 10 secondary drops per collision. Schremb et al. 2017 observed on the order of tens of drops per collisions for impacts on an elevated ice surface. Finally, Rozhkov et al. 2002 observed hundreds of drops for impacts on steel disks…"